# Proinsulin misfolding is an early event in the progression to type 2 diabetes

Anoop Arunagiri[1], Leena Haataja[1], Anita Pottekat[2], Fawnnie Pamenan[1], Soohyun Kim[3], Lori M Zeltser[4], Adrienne W Paton[5], James C Paton[5], Billy Tsai[6], Pamela Itkin-Ansari[7], Randal J Kaufman[2]*, Ming Liu[1,8], Peter Arvan[1]*

[1]Division of Metabolism, Endocrinology & Diabetes, University of Michigan Medical School, Ann Arbor, United States; [2]Degenerative Diseases Program, Sanford Burnham Prebys Medical Discovery Institute, La Jolla, United States; [3]Department of Biomedical Science and Technology, Konkuk University, Gwangjin-gu, Republic of Korea; [4]Department of Pathology and Cell Biology, Naomi Berrie Diabetes Center, Columbia University, New York, United States; [5]Research Centre for Infectious Diseases, Department of Molecular and Biomedical Science, University of Adelaide, Adelaide, Australia; [6]Department of Cell and Developmental Biology, University of Michigan, Ann Arbor, United States; [7]Development, Aging and Regeneration Program, Sanford Burnham Prebys Medical Discovery Institute, La Jolla, United States; [8]Department of Endocrinology and Metabolism, Tianjin Medical University, Tianjin, China

*For correspondence:
rkaufman@sbpdiscovery.org (RJK);
parvan@umich.edu (PA)

Competing interests: The authors declare that no competing interests exist.

**Abstract** Biosynthesis of insulin – critical to metabolic homeostasis – begins with folding of the proinsulin precursor, including formation of three evolutionarily conserved intramolecular disulfide bonds. Remarkably, normal pancreatic islets contain a subset of proinsulin molecules bearing at least one free cysteine thiol. In human (or rodent) islets with a perturbed endoplasmic reticulum folding environment, non-native proinsulin enters intermolecular disulfide-linked complexes. In genetically obese mice with otherwise wild-type islets, disulfide-linked complexes of proinsulin are more abundant, and leptin receptor-deficient mice, the further increase of such complexes tracks with the onset of islet insulin deficiency and diabetes. Proinsulin-Cys(B19) and Cys(A20) are necessary and sufficient for the formation of proinsulin disulfide-linked complexes; indeed, proinsulin Cys(B19)-Cys(B19) covalent homodimers resist reductive dissociation, highlighting a structural basis for aberrant proinsulin complex formation. We conclude that increased proinsulin misfolding via disulfide-linked complexes is an early event associated with prediabetes that worsens with ß-cell dysfunction in type two diabetes.
DOI: https://doi.org/10.7554/eLife.44532.001

## Introduction

The progression of β-cell dysfunction with endoplasmic reticulum (ER) stress in prediabetes, to ß-cell failure with full-blown type two diabetes (T2D), has long been the subject of intensive investigation (*Fonseca et al., 2011*; *Rabhi et al., 2014*). What is observed is that a chronic increase in insulin secretory demand over time leads to depletion of ß-cell secretory granule reserves despite a dramatic compensatory increase in secretory pathway activity (*Alarcon et al., 2016*; *Rustenbeck, 2002*). By electron microscopy, pancreatic sections of human subjects with T2D show a marked increase of ER volume density in β-cells with an induction of ER stress markers triggered by high glucose exposure (*Marchetti et al., 2007*). Similar indicators of ER stress have been established in the β-cells of animal models (*Yang et al., 2016*), such as leptin receptor-deficient diabetic mice [known as LepR*db/*

**eLife digest** Our body fine-tunes the amount of sugar in our blood thanks to specialized 'beta cells' in the pancreas, which can release a hormone called insulin. To produce insulin, the beta cells first need to build an early version of the molecule – known as proinsulin – inside a cellular compartment called the endoplasmic reticulum. This process involves the formation of internal staples that keep the molecule of proinsulin folded correctly.

Individuals developing type 2 diabetes have spikes of sugar in their blood, and so their bodies often respond by trying to make large amounts of insulin. After a while, the beta cells can fail to keep up, which brings on the full-blown disease.

However, scientists have discovered that early in type 2 diabetes, the endoplasmic reticulum of beta cells can already show signs of stress; yet, the exact causes of this early damage are still unknown. To investigate this, Arunagiri et al. looked into whether proinsulin folds correctly during the earliest stages of type 2 diabetes. Biochemical experiments showed that even healthy beta cells contained some misfolded proinsulin molecules, where the molecular staples that should fold proinsulin internally were instead abnormally linking proinsulin molecules together. Further work revealed that the misfolded proinsulin was accumulating inside the endoplasmic reticulum. Finally, obese mice that were in the earliest stages of type 2 diabetes had the highest levels of abnormal proinsulin in their beta cells.

Overall, the work by Arunagiri et al. suggests that large amounts of proinsulin molecules stapling themselves to each other in the endoplasmic reticulum of beta cells could be an early hallmark of the disease, and could make it get worse. A separate study by Jang et al. also shows that a protein that limits the misfolding of proinsulin is key to maintain successful insulin production in animals eating a Western-style, high fat diet.

Hundreds of millions of people around the world have type 2 diabetes, and this number is rising quickly. Detecting and then fixing early problems associated with the condition may help to stop the disease in its track.

DOI: https://doi.org/10.7554/eLife.44532.002

*db* (*Diani et al., 1984*; *Laybutt et al., 2007*; *Like and Chick, 1970*) that develop insulin resistance progressing to T2D, which is linked to overeating. Hypersynthesis of proinsulin (*Arunagiri et al., 2018*; *Back et al., 2009*) is a condition proposed to increase proinsulin misfolding (*Liu et al., 2005*; *Scheuner et al., 2005*) — which can promote ER stress with abnormal ß-cell ER expansion — whereas suppression of proinsulin protein synthesis actually alleviates ß-cell ER stress (*Szabat et al., 2016*).

Insulin-deficiency caused directly by proinsulin misfolding has been proved unequivocally in an autosomal-dominant form of diabetes known as Mutant *INS*-gene-induced Diabetes of Youth (MIDY) (*Liu et al., 2010b*), which occurs in patients bearing one mutant and one wild-type *INS* allele (*Liu et al., 2015*; *Støy et al., 2010*). The disease in humans is pathogenetically identical to that seen in the mutant *Akita* diabetic mouse (*Izumi et al., 2003*) or Munich MIDY Pig (*Blutke et al., 2017*) – which are animals expressing one mutant *INS* allele encoding proinsulin-C(A7)Y that is quantitatively misfolded due to an inability to form the Cys(B7)-Cys(A7) disulfide bond. Ordinarily the expression of only one WT *INS* allele would be sufficient to avoid diabetes, but *Akita* mice develop diabetes despite expressing three alleles encoding WT proinsulin in addition to the one encoding mutant pro-insulin (*Liu et al., 2010b*). Both preclinical and clinical data prove that in MIDY, it is the expression of misfolded proinsulin that triggers diabetes; yet MIDY is a rare disease. Of far broader significance is the β-cell failure that accompanies 'garden variety' T2D without *INS* mutations, and though the molecular pathogenesis of insulin deficiency in this condition remains murky (*Halban et al., 2014*), β-cell ER stress is a recognized part of the disease. It has been suggested that β-cells compensate for insulin resistance by increasing insulin production that may eventually overwhelm the ER capacity for efficient protein folding, thereby provoking β-cell ER stress (*Back and Kaufman, 2012*; *Eizirik et al., 2008*; *Herbert and Laybutt, 2016*; *Papa, 2012*; *Rabhi et al., 2014*; *Volchuk and Ron, 2010*). However, in the absence of *INS* gene mutations, it has not been established the extent to which proinsulin misfolding is present in the early triggering stages of T2D, including prediabetes and mild

dysglycemia — prior to more obvious islet failure including β-cell degranulation and dedifferentiation (*Accili et al., 2016*; *Kahn, 1998*; *Kahn et al., 2009*) that occurs in both human islets (*Cinti et al., 2016*) and rodent islets (*Ishida et al., 2017*). In this study, we have exploited several independent lines of evidence to establish the presence of aberrant disulfide-linked proinsulin complexes in the β-cells of human islets and model systems, in states that alter the ER folding environment, and in T2D progression prior to onset of β-cell dedifferentiation (*Bensellam et al., 2018*) or death (*Eizirik and Millard, 2014*; *Kanekura et al., 2015*; *Marchetti et al., 2012*; *Papa, 2012*).

## Results

### Proinsulin in the ER has reactive cysteine thiols and is predisposed to aberrant Disulfide-Linked complex formation

Both murine islets and the INS1 (rat) pancreatic ß-cell line cells secrete successfully-folded proinsulin in addition to processed insulin. Native proinsulin folding requires formation of Cys(B7)-Cys(A7), Cys (B19)-Cys(A20) and Cys(A6)-Cys(A11) disulfide pairs (*Haataja et al., 2016*). One way to detect improperly folded wild-type proinsulin in pancreatic β-cells is to look for the possible presence of unpaired Cys residues. Alkylation of proinsulin Cys residues with 4-acetamido-4'-maleimidyl-stilbene-2,2'-disulfonate (AMS) adds 0.5 kD of molecular mass for each cysteine modified, shifting proinsulin from its normal molecular mass. As examined by immunoblotting with anti-proinsulin antibody, no modification by AMS could be detected in secreted recombinant human proinsulin or proinsulin from rodent islets, or INS cells (e.g., *Figure 1—figure supplement 1A*). Remarkably, however, alkylation of intracellular proinsulin with AMS in human islets caused a decrease in unmodified proinsulin accompanied by the appearance of proinsulin alkylated on at least one cysteine thiol (*Figure 1A*). Alkylation of intracellular proinsulin was also observed in rodent islets (*Figure 1—figure supplement 1B*). The presence of a free thiol in a significant subpopulation of proinsulin molecules can lead to inappropriate intermolecular disulfide attack on neighboring proinsulin molecules (*Cunningham et al., 2017*; *Liu et al., 2010a*; *Liu et al., 2007*; *Wang et al., 1999*).

Antibodies have been described that recognize misfolded proinsulin molecules bearing intermolecular disulfide bonds (*Lee et al., 2016*; *Wang et al., 2011*). Indeed, WT proinsulin is predisposed to misfolding (*Haataja et al., 2016*). Proinsulin misfolding has not been demonstrated to be exacerbated in ß-cells deficient for Ire1 or ATF6, but it has been repeatedly found to be exacerbated in ß-cells with dysfunctional PERK (caused either by gene knockout, dominant-negative mutant, or specific chemical inhibitor) — leading to what has been described as the 'proinsulin-impacted-ER' phenotype (*Gupta et al., 2010*; *Harding et al., 2012*; *Scheuner et al., 2005*). We performed immunoblotting of nonreducing SDS-PAGE samples with a monoclonal antibody that recognizes rodent proinsulin but not insulin (mAb CCI-17, *see* Materials and methods), with the intent to identify intermolecular disulfide-linked proinsulin complexes. Immunoblotting of either untreated ß-cells or those treated with vehicle alone detected proinsulin monomers and disulfide-linked dimers (*Figure 1B left*) the latter of which are, by definition, non-native. Moreover, whereas inhibition of PERK dramatically increased the intracellular accumulation of proinsulin and shifted its distribution to the classic impacted-ER phenotype (*Figure 1C*) (*Gupta et al., 2010*), a remarkable ladder of higher molecular mass bands appeared upon Western blotting with mAb anti-proinsulin after nonreducing SDS-PAGE (*Figure 1B left*). Despite this increase of intracellular proinsulin, secretion of proinsulin to the medium was not increased; furthermore, by nonreducing SDS-PAGE, secreted proinsulin was recovered exclusively as the monomeric form (*Figure 1B right*), demonstrating efficient quality control in the secretory pathway. The intracellular accumulation of a ladder of proinsulin immunoreactive species upon nonreducing SDS-PAGE was not detected by conventional Western blotting with polyclonal anti-insulin, despite that this antibody identifies both proinsulin and proinsulin conversion intermediates (*Figure 1D left*). However, dimers and higher order complexes were detected by Western blotting of the identical samples with mAb anti-proinsulin (*Figure 1D right*), and all such higher bands, comprising the majority of intracellular proinsulin molecules, collapsed to monomers upon reduction of disulfide bonds (*Figure 1—figure supplement 1C*).

To confirm that the ladder of Western blotted bands detected by nonreducing SDS-PAGE specifically reflects disulfide-linked complexes of proinsulin, a similarly-generated sample was analyzed on a second-dimensional reducing SDS-PAGE, which demonstrated that nearly all bands in the

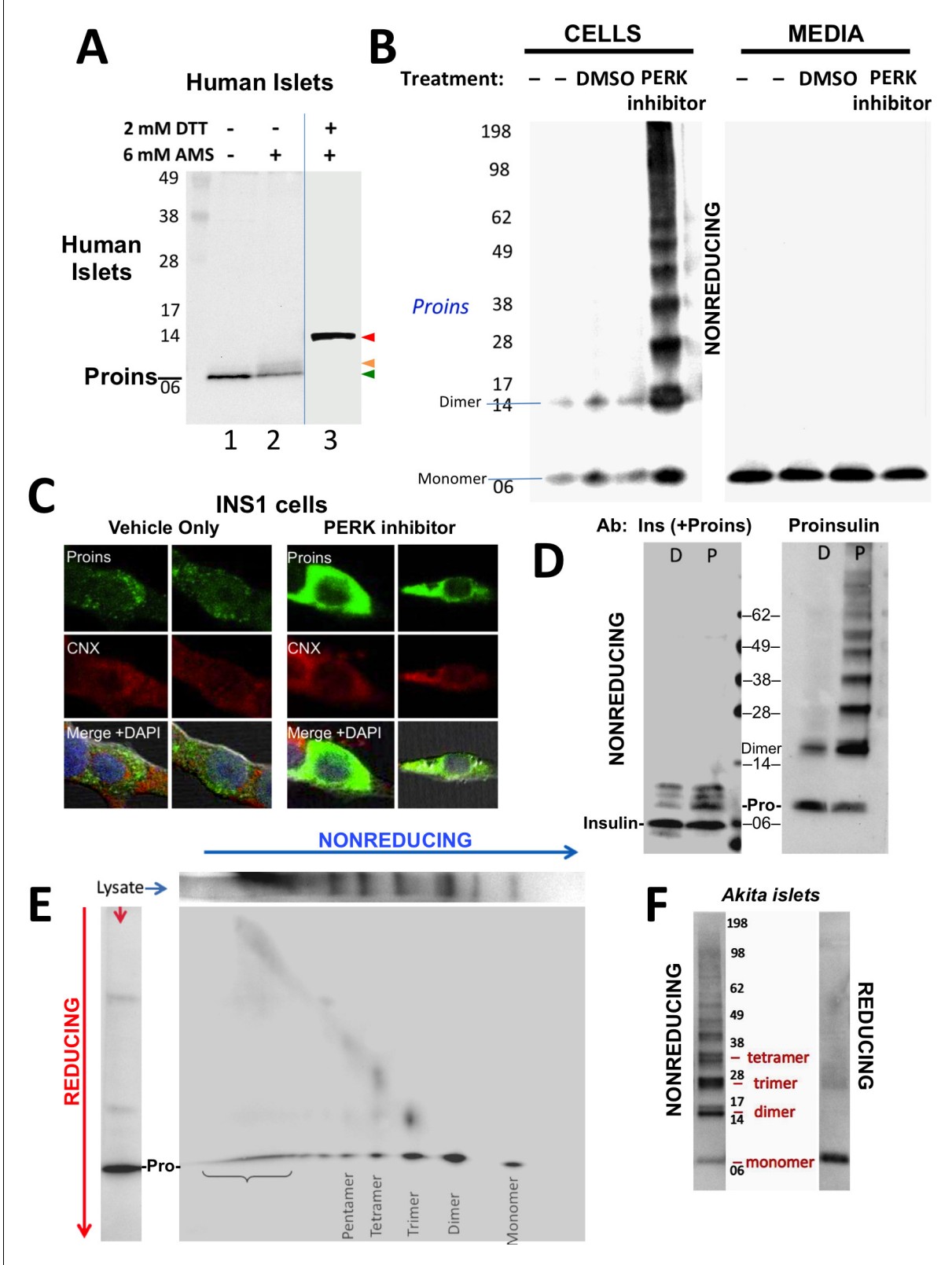

**Figure 1.** Detection of improperly folded proinsulin. (A) Non-diabetic human pancreatic islets were lysed in RIPA buffer and divided into three equal parts, one of which was (partially) pre-reduced by boiling in the presence of 2 mM DTT (*lane 3*). This and a non-pre-reduced sample (*lane 2*) underwent alkylation with 6 mM AMS. A third sample was neither pre-reduced nor alkylated (*lane 1*). All samples were incubated for 1 hr at 37°C and finally resolved by SDS-PAGE under reducing conditions (200 mM DTT), electrotransfer to nitrocellulose, and immunoblotting for human proinsulin with mAb

*Figure 1 continued on next page*

*Figure 1 continued*

20G11. The red arrowhead: proinsulin species with >1 alkylated Cys (*upper band*); beige arrowhead: at least one alkylated Cys (*middle band*); green arrowhead: no free Cys (*bottom band*). (B) Cell lysate (*left*) and overnight secretion (*right*) from untreated INS1E cells (–) or those treated with vehicle alone (DMSO) or PERK inhibitor (GSK2656157, 2 μM) were analyzed by nonreducing SDS-PAGE, electrotransfer to nitrocellulose, and immunoblotting for rodent proinsulin (mAb CCI-17). The positions of molecular mass markers are noted. (C). INS1E cells were treated overnight with vehicle (DMSO) or PERK inhibitor before formaldehyde fixation, permeabilization, and indirect immunofluorescence with mAb anti-proinsulin (GS-9A8, green) and rabbit anti-calnexin (red), with appropriate secondary antibodies. (D) INS1E cells were treated with DMSO (lane marked 'D') or PERK inhibitor (lane marked 'P'). The cells were lysed and resolved in duplicate by nonreducing SDS-PAGE. The final gel was treated with 25 mM DTT for 10 min at 25°C, electrotransferred to nitrocellulose, and then immunoblotted with guinea pig anti-insulin that cross-reacts with proinsulin ('Pro') and conversion intermediates (*left panel*) or anti-proinsulin (CCI-17, *right panel*). The positions of molecular mass markers are noted. (E) INS1E cells treated with PERK inhibitor as in panel B) were lysed and resolved by a first dimensional nonreducing SDS-PAGE (shown horizontally, *at top*) and then in a second dimensional reducing SDS-PAGE (shown vertically, *at left*). The 2D gel was electrotransferred to nitrocellulose and immunoblotted for rodent proinsulin (mAb CCI-17). The bracket indicates high molecular weight proinsulin-containing complexes. (F) Islets from *Akita* male mouse were lysed in RIPA buffer and analyzed by anti-proinsulin immunoblotting (mAb CCI-17) under nonreducing (*left*) or reducing conditions (*right*). The positions of molecular mass markers are noted.

DOI: https://doi.org/10.7554/eLife.44532.003

The following figure supplement is available for figure 1:

**Figure supplement 1.** Proper and improper disulfide bond formation in proinsulin.

DOI: https://doi.org/10.7554/eLife.44532.004

nonreduced ladder contained proinsulin, with molecular masses expected of disulfide-linked complexes as simple as homodimers, homotrimers, homotetramers, and homopentamers, as well as higher order complexes (*Figure 1E*). Quantitatively these complexes comprised 87% of all recovered proinsulin. This is very similar to what has been reported for the islets of *Akita* mice that express misfolded proinsulin from one mutant allele that entraps wild-type proinsulin expressed from three wild-type alleles (*Izumi et al., 2003*). Indeed, Western blotting of nonreduced lysates of *Akita* mouse islets with anti-proinsulin demonstrated that the majority of proinsulin was recovered in aberrant disulfide-linked proinsulin complexes (*Figure 1F*), which were converted to monomeric proinsulin upon SDS-PAGE under reducing conditions.

To confirm that even in the absence of *INS* gene mutations, formation of disulfide-linked proinsulin complexes constitutes improper proinsulin folding, we exposed INS1 pancreatic beta cells to PERK inhibitor for times up to 20 hr. With increasing time of exposure, the majority of intracellular proinsulin accumulated in disulfide-linked complexes in the ß-cells (*Figure 2A second panel*), and this increase was ultimately accompanied by decreased insulin content (*third panel*) as well as decreased proinsulin secretion (*final panel*). These observations are consistent with ER quality control limiting the ER export of misfolded proinsulin resulting in diminished delivery to post-Golgi sites (including the extracellular medium, also noted in *Figure 1B*). Significantly, treatment of intracellular proinsulin dimers and larger complexes with AMS, followed by nonreducing SDS-PAGE, revealed that each of the oligomeric species in the ladder of bands shifted up after alkylation, indicating that each of these forms still bears at least one unpaired cysteine thiol (*Figure 2B*). These data provide a rationale for the propagation of disulfide-linked dimers into trimers, trimers into tetramers, etc. We found that pretreatment of cells with NEM before (and during) cell lysis results in the detection of dramatically less of the ladder of disulfide-linked proinsulin oligomers (*Figure 2C*) — and additionally, NEM pretreatment favors an increase in the detection of higher molecular weight proinsulin-containing complexes. Nevertheless, the relative abundance of the distinct disulfide-linked oligomeric species of proinsulin was essentially unchanged in cells that were either alkylated in situ with 20 mM N-ethyl maleimide prior to cell lysis, or not (*Figure 2C*). These data render it unlikely that the detection of aberrant disulfide-linked complexes of nonmutant proinsulin is an artifact of cell lysis conditions.

## Aberrant Disulfide-Linked complex formation of proinsulin in human islets

To explore whether human proinsulin (*Figure 1A*) is similarly predisposed to the aberrant disulfide-linked complex formation that was detected in rodent ß-cells, we first expressed recombinant human proinsulin in heterologous cells, followed by nonreducing SDS-PAGE and immunoblotting with a

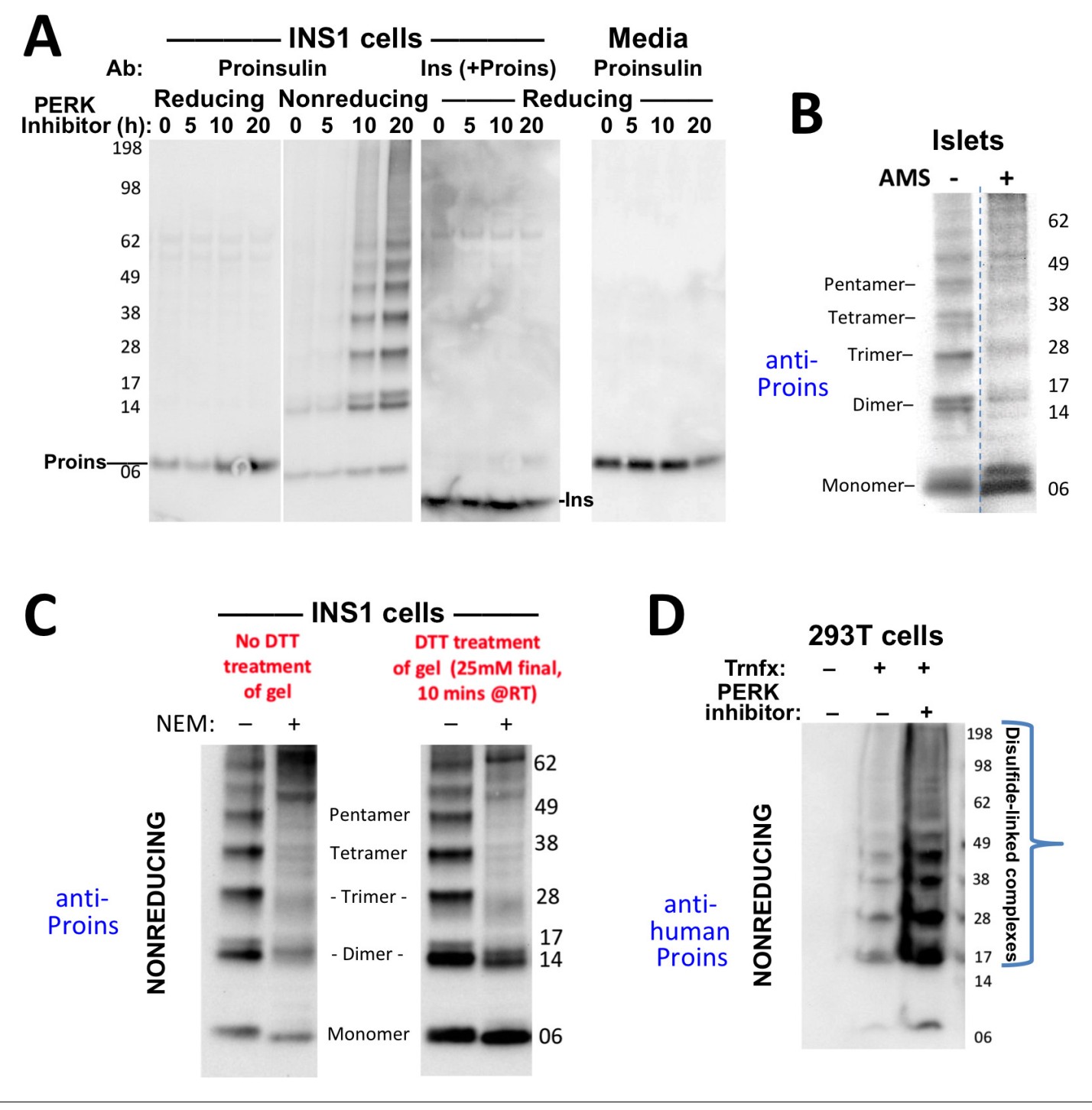

**Figure 2.** Formation of proinsulin disulfide-linked complexes. (A) INS1E cells were incubated for 20 hr in culture medium; the last 0, 5, 10, or 20 hr of this incubation included PERK inhibitor as indicated. At the end of the 20 hr, the media were collected and the cells were lysed; samples were resolved by nonreducing or reducing SDS-PAGE and electrotransferred to nitrocellulose. Panels 1, 2, and four were immunoblotted with mAb anti-proinsulin (CCI-17); panel three was immunoblotted with guinea pig anti-insulin. The positions of molecular mass markers are noted. (B) Murine pancreatic islets treated overnight with PERK inhibitor were lysed in SDS gel sample buffer under nonreducing conditions, and divided into two portions. One portion of lysate was incubated with 6 mM AMS for 1 hr, and then both portions were resolved by nonreducing SDS-PAGE, electrotransferred to nitrocellulose, and immunoblotted with mAb anti-proinsulin (CCI-17). The availability of free thiols results in proinsulin bands shifting in the AMS-treated lysate (*right*) compared to the untreated one (*left*). The positions of molecular mass markers are noted. (C). INS1E cells treated with PERK inhibitor overnight were washed with ice cold PBS either lacking or containing 20 mM NEM and lysed in the absence (lane 1) or presence of 2 mM NEM (lane 2). These samples
*Figure 2 continued on next page*

*Figure 2 continued*

were run on two halves of the same nonreducing SDS-PAGE; the right half-gel was incubated for 10 min with 25 mM DTT, while the left half-gel remained untreated. Both halves were then electrotransferred and immunoblotted with mAb anti-proinsulin. DTT treatment of the gel increased the signal strength of proinsulin monomers (less so for dimers). Conversely, alkylation greatly decreased the signal strength of disulfide-linked proinsulin oligomers but yielded a similar ratio of distinct proinsulin oligomeric species to that seen without alkylation. Molecular mass markers are indicated. (**D**) 293 T cells were either mock-transfected or transfected ('Trnfx') to express recombinant human proinsulin. Cells were treated overnight with vehicle (–) or PERK inhibitor before lysis, nonreducing SDS-PAGE, electrotransfer to nitrocellulose, and immunoblotting with mAb anti-human proinsulin (20G11). The positions of molecular mass markers are noted.

DOI: https://doi.org/10.7554/eLife.44532.005

The following figure supplement is available for figure 2:

**Figure supplement 1.** Proinsulin disulfide-linked complexes in human islets.

DOI: https://doi.org/10.7554/eLife.44532.006

monoclonal antibody that reacts exclusively with the amino-terminal region of human C-peptide (mAb 20G11, see Materials and methods). Disulfide-linked proinsulin dimers along with a ladder of higher-order complexes already represented the majority of recombinant proinsulin species, and these forms accumulated to a dramatically higher level in cells treated overnight with PERK inhibitor (*Figure 2D*).

We then looked for the presence of such disulfide-linked dimers and higher order complexes formed by endogenous proinsulin in preparations of human islets from unrelated donors without a history of T2D. In several preparations, disulfide-linked dimers were prominent, along with a lesser abundance of higher-order complexes (*Figure 3A right panel*). In human islets obtained from other individual donors, disulfide-linked dimers were less prominent (*Figure 2—figure supplement 1* and *Figure 3—figure supplement 1A*). Overnight treatment of human islets with PERK inhibitor resulted in a further increase in higher order disulfide-linked complexes (*Figure 3A right panel*, and *Figure 3—figure supplement 1A*) and an increase in total islet proinsulin as revealed by reducing SDS-PAGE (*Figure 3A left panel*).

These data indicate that aberrant disulfide-linked proinsulin dimers and higher complexes exist in human islets and their abundance responds to changes in environmental conditions within the ER. The immunodetection of both disulfide-linked dimers and higher-order complexes was blocked when pure human C-peptide competitor was included in the immunoblotting protocol (*Figure 3B*), demonstrating specificity.

PERK inhibitor has a rapid onset of action on its target, but it appears that the impact of this inhibition to globally alter the proinsulin folding environment of the ER (*Wang et al., 2013*) may take a half-day or more (*Figure 2A*). More immediately, nascent proinsulin binds the hsp70 family member, BiP (*Liu et al., 2005*; *Scheuner et al., 2005*); thus, for a more direct perturbation, we exposed islets to the bacterial SubAB protease that is endocytosed and retrieved to the ER lumen where it cleaves the ER chaperone BiP within a matter of a few hours or less (*Paton et al., 2006*). In mouse islets, BiP was $\geq$90% destroyed within 2 hr after SubAB addition (*Figure 3C upper left*), and immunoblotting revealed that this treatment shifted the majority of proinsulin into larger disulfide-linked complexes (*Figure 3C lower right*). Similarly, in a rodent ß-cell line, improperly folded proinsulin began to migrate increasingly in disulfide-linked complexes 80 min after SubAB addition when only a portion of intracellular BiP had been destroyed (*Figure 3—figure supplement 1B*). BiP cleavage was also confirmed by appearance of the ~28 kD C-terminal BiP cleavage fragment (*Figure 3—figure supplement 1C*). In the islets of humans not known to have T2D, improperly folded disulfide-linked proinsulin dimers were already apparent prior to SubAB addition, and an increase in larger-sized covalent proinsulin complexes was apparent at 4 hr of SubAB treatment (*Figure 3D lower panel*). These data demonstrate an acute increase in improper folding of human proinsulin, without any *INS* gene mutation, under conditions of BiP deficiency within the ß-cell ER luminal environment.

The hsp90 family member of the ER, GRP94, also impacts on proinsulin handling in ß-cells, especially resulting in aberrant post-ER processing with markedly abnormal appearing secretory granules (*Ghiasi et al., 2019*); however, treatment of ß-cells with GRP94 inhibitor (PU-WS13, 20 µM) even for 24 hr did not promote proinsulin disulfide-linked complex formation, and did not exacerbate the proinsulin disulfide-linked complex formation that was triggered by an acute (3 hr) loss of BiP (*Figure 4A*). Immunofluorescence microscopy demonstrated that after SubAB treatment of ß-cells,

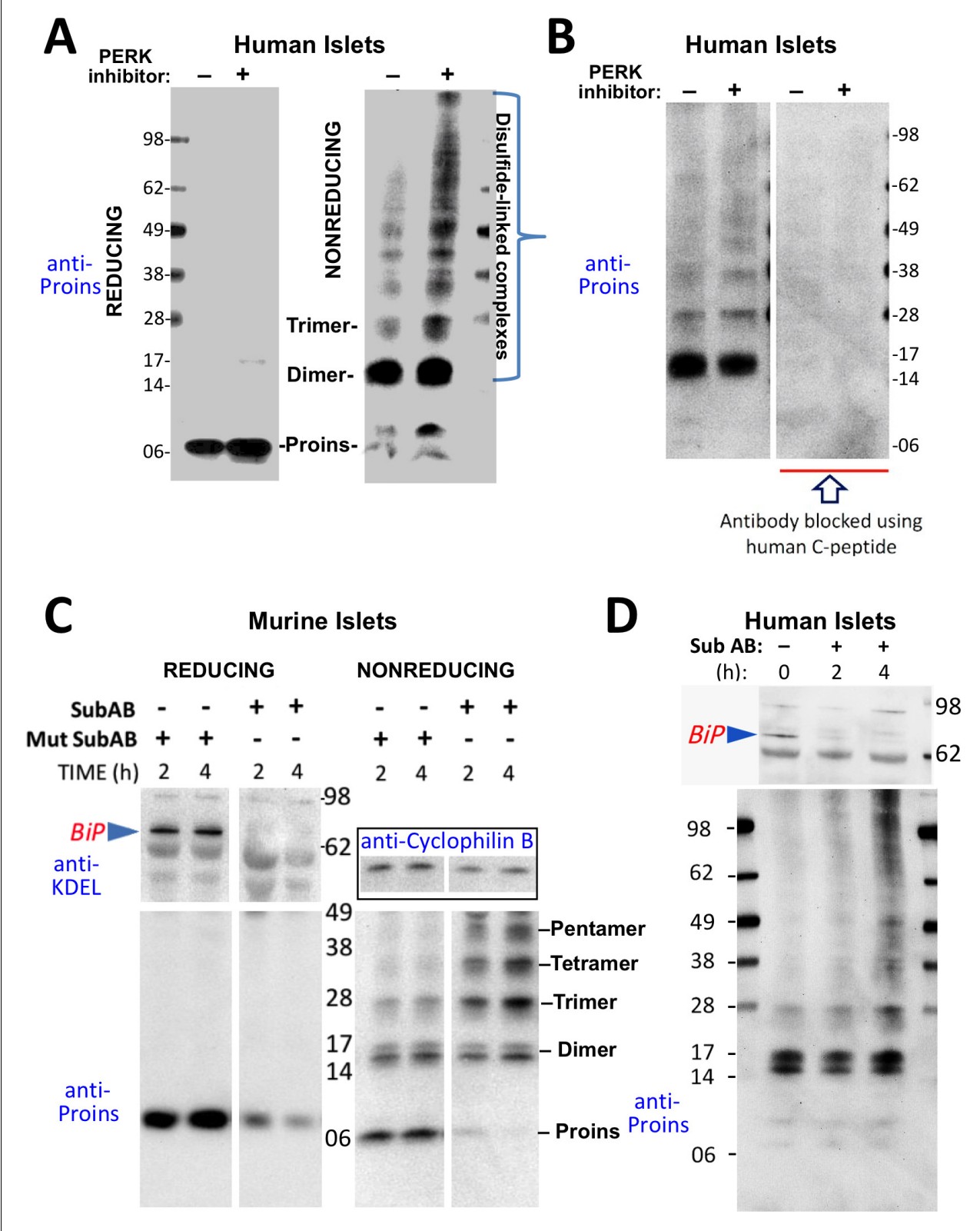

**Figure 3.** Misfolding of proinsulin in human (and rodent) pancreatic islets. (A) Islets from humans not known to be diabetic (Prodo Labs) were treated overnight with vehicle or PERK inhibitor before lysis, reducing or nonreducing SDS-PAGE, electrotransfer to nitrocellulose, and immunoblotting for human proinsulin (mAb 20G11). The positions of molecular mass markers are noted. (B) Human islet lysates were immunoblotted with mAb anti-proinsulin (20G11)±purified human C-peptide as a blocking/competitor peptide (molecular mass markers indicated). (C) Mouse pancreatic islets were
*Figure 3 continued on next page*

*Figure 3 continued*

isolated and maintained in an overnight recovery medium including 11.1 mM glucose. The islets were then incubated with active SubAB (1 µg/mL final) or the same concentration of inactive mutant SubA$_{A272}$B for 2 hr or 4 hr, as indicated. At each time point the islets were lysed and analyzed by reducing or nonreducing SDS-PAGE and immunoblotting with anti-KDEL (recognizing multiple ER resident proteins including BiP, as indicated), mAb anti-proinsulin (CCI-17, *lower panels*), and anti-cyclophilin B (loading control, *boxed*). Molecular mass markers are noted. (D) Islets from humans not known to be diabetic (obtained from the IIDP) were incubated with active SubAB (1.5 µg/mL final) for 0 hr, 2 hr or 4 hr, as indicated. At each time point the islets were lysed and analyzed by nonreducing SDS-PAGE and immunoblotting with anti-KDEL (*upper panel*) and mAb anti-proinsulin (20G11, *lower panel*). The positions of molecular mass markers are noted.
DOI: https://doi.org/10.7554/eLife.44532.007

The following figure supplement is available for figure 3:

**Figure supplement 1.** Proinsulin misfoding induced by perturbation of the ER folding environment.
DOI: https://doi.org/10.7554/eLife.44532.008

the intracellular distribution of proinsulin shifted from its usual predominant juxtanuclear localization [in the Golgi region from which newly-made insulin granules emerge (*Haataja et al., 2013*) to a co-localization with the ER marker, calnexin (*Figure 4B*). Altogether, these data strongly indicate that proinsulin disulfide-linked complexes are misfolded and are retained within the ER compartment.

## The dynamics of proinsulin folding and intracellular distribution in a T2D model

The predominant juxtanuclear Golgi-regional distribution of proinsulin is a feature of normal rodent islets (*Orci et al., 1985*) (*Figure 4C first set of panels*). Within the first 4–5 months of life, leptin receptor mutant LepR$^{db/db}$ mice in a C57BL/6 background become severely diabetic and develop a paucity of islet ß-cells immunostainable for mature insulin — indeed, the fraction of islet ß-cells that exhibit little or no insulin immunostaining is known to increase by more than 5-fold compared to the islets of age-matched control mice (*Ishida et al., 2017*). However, we observed that proinsulin-positive immunostaining amongst the cells in these diabetic islets was more widespread and the intracellular distribution of proinsulin within these islet ß-cells was no longer as concentrated in the Golgi region (*Figure 4C second and third set of panels*) with increased localization in the ER (*Figure 4—figure supplement 1*). It has been established that limiting food intake of these animals can substantially increase the percentage of strongly insulin-immunopositive ß-cells per islet (*Ishida et al., 2017*). Indeed, upon fasting overnight, the islets of diabetic LepR$^{db/db}$ mice exhibited a more robust immunostaining of mature insulin (*Figure 4C fourth and fifth set of panels*). Additionally, after fasting, there was a shift in intracellular proinsulin back towards a juxtanuclear distribution (*Figure 4C fourth and fifth set of panels*), consistent with previous reports (*Alarcon et al., 2016*). The data of *Figure 4* strongly suggest that in *ad lib* fed LepR$^{db/db}$ diabetic mice, initial insulin depletion does not preclude ongoing proinsulin expression; moreover, a shift of proinsulin distribution towards an ER-like pattern (*Figure 4—figure supplement 1*) may suggest a stressed state that may be preventable and to some extent reversible by limiting ß-cell secretory stimulation (i.e., 'ß-cell rest') (*Ishida et al., 2017*).

We wondered if the ER-like distribution of intracellular proinsulin in LepR$^{db/db}$ mice might be explained by (nonmutant) proinsulin folding exceeding the ER folding capacity of these ß-cells, resulting in the accumulation of improperly folded species. With this in mind, we performed proinsulin immunoblotting after nonreducing and reducing SDS-PAGE in LepR$^{db/db}$ islet lysates, with islet lysates from wild-type mice, or those bearing only one of 4 functional *Ins* alleles, as controls. Under reducing conditions, >99% of proinsulin was monomeric (and insulin was recovered as the reduced insulin B-chain, *Figure 4D right*). We examined proinsulin in the islet lysates of 5 additional prediabetic LepR$^{db/db}$ mice — all were similar to the islets of two animals shown in *Figures 4D* (4 and 6 weeks of age with random blood glucose levels of 101 and 118 mg/dL, respectively) containing a large steady state level of proinsulin (seen upon Western blotting of reducing SDS-PAGE) with >90% of molecules entangled in disulfide-linked dimers and higher-order complexes (seen by Western blotting upon nonreducing SDS-PAGE, *Figure 4D*). Accumulation of misfolded proinsulin in the ER of LepR$^{db/db}$ ß-cells was always present before the onset of diabetes. These bands are indeed proinsulin, as they were absent from islets of mice depleted of three of four *Ins* alleles; *Figure 4D* lane 3).

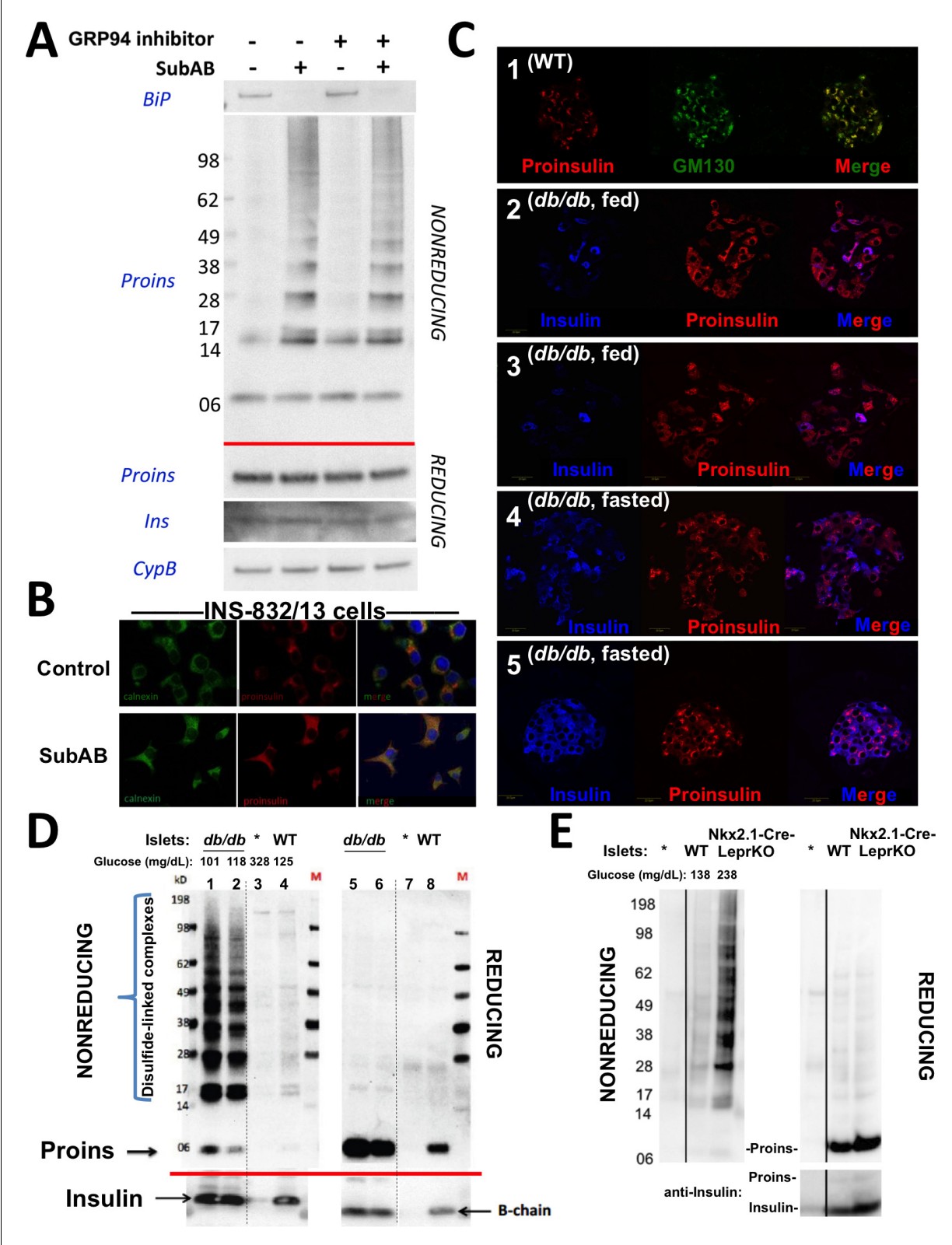

**Figure 4.** Improper proinsulin folding from pharmacological or physiological alteration of the ß-cell ER folding environment. (**A**) INS1E cells were incubated for 24 hr ± 20 μM GRP94 inhibitor (PU-WS13). During the last 3 hr, SubAB was added where indicated. Cell lysates were analyzed by immunoblotting for BiP (*top panel*), and mAb anti-proinsulin (CCI-17) under nonreducing conditions (*above red line*) or with anti-proinsulin (Proins), anti-insulin (Ins), and anti-cyclophilin B (CypB, loading control) under reducing conditions (*below red line*). The positions of molecular mass markers are
*Figure 4 continued on next page*

*Figure 4 continued*

noted. (B) INS832/13 incubated ±active SubAB (1.5 µg/mL, 4 hr) were processed for immunofluorescence with rabbit anti-calnexin (green) or mAb anti-proinsulin (red). (C) Sections of wild-type or LepR$^{db/db}$ mouse pancreas (C57BL/6) were de-paraffinized and prepared for indirect immunofluorescence. 1) Wild-type islets immunostained with mAb anti-proinsulin (CCI-17, in red), or the Golgi complex labeled with mab anti-GM130 (in green). 2–5) LepR$^{db/db}$ mice with random blood glucose >500 mg/dL, as follows. 2 and 3) Mice fed *ad lib*; immunostained for insulin (blue) and mAb anti-proinsulin (CCI-17, red). 4 and 5) Mice fasted overnight, and immunostained as above. The third panel in each case is a merged image of the single-channel fluorescence. (D) Isolated islets from young male LepR$^{db/db}$ mice (lanes 1, 2, 5 and 6; *random blood glucose values shown above*) or wild-type C56BL/6 ('WT', lanes 4, 8) or a high-fat fed *Ins1$^{+/-}$,Ins2$^{-/-}$* male (lanes 3 and 7, marked with asterisk) were lysed in RIPA buffer and analyzed by nonreducing or reducing SDS-PAGE and immunoblotting with mAb anti-proinsulin (CCI-17, *above red line*) or guinea pig anti-insulin (*below red line*). The positions of molecular mass markers are noted. (E) Isolated islets from WT and Nkx2.1-Cre-mediated LepR-KO mice (*random blood glucose values shown above*) were lysed in RIPA buffer and analyzed under nonreducing or reducing conditions as in panel D. Molecular mass markers are noted. Asterisk denotes lysate from *Ins1$^{+/-}$, Ins2$^{-/-}$* islets as in panel D. The positions of molecular mass markers are noted. Islet lysates from WT and Nkx2.1-Cre-mediated LepR KO mice were also immunoblotted with guinea pig anti-insulin (*bottom right*) that weakly cross reacts with proinsulin (Proins).
DOI: https://doi.org/10.7554/eLife.44532.009

The following source data and figure supplements are available for figure 4:

**Figure supplement 1.** Intracellular proinsulin distribution in the LepR$^{db/db}$ mouse.

DOI: https://doi.org/10.7554/eLife.44532.010

**Figure supplement 1—source data 1.**

DOI: https://doi.org/10.7554/eLife.44532.011

Importantly, improperly folded proinsulin was also detectable in WT mouse islets, albeit at a much lower abundance (*Figure 4D* lane 4).

Because leptin receptors might have direct actions on ß-cells (*Marroquí et al., 2012*), we also examined islets from LepR-Nkx2.1 KO mice that are deficient for leptin receptor in the hypothalamus but not in the islets (*Ring and Zeltser, 2010*). Here too, pancreatic islets from these hyperphagic, obese, glucose intolerant animals demonstrated markedly increased abundance of disulfide-linked proinsulin complexes with >90% of proinsulin entangled in such complexes (*Figure 4E*). These data indicate that an increase in improper proinsulin folding in leptin receptor-deficient animals is secondary to a prediabetic state and unrelated to leptin receptor function in ß-cells. Crucially, these data also demonstrate that improper proinsulin folding is already detectable at a time that has been associated with ER stress response (*Herbert and Laybutt, 2016*) but before onset of chronic hyperglycemia associated with pancreatic insulin deficiency that has been attributed to a loss of functional ß-cell mass [regardless of whether this is due to de-differentiation (*Bensellam et al., 2018*) or ß-cell death (*Eizirik and Millard, 2014*; *Kanekura et al., 2015*; *Marchetti et al., 2012*; *Papa, 2012*) or both].

Based on the work of *Laybutt et al. (2007)*, we examined islet protein content of p58$^{ipk}$ (encoded by DNAJC3) as a reliable marker of islet ER stress response, and found that the levels of this ER co-chaperone of BiP began to increase in *ad lib* fed LepR$^{db/db}$ mice even before the animals have reached a random blood glucose of 160 mg/dL (*Figure 5A*). Islet p58$^{ipk}$ protein levels also tended to be elevated in LepR-Nkx2.1 KO mice (*Figure 5B*).

In the islets of young, normoglycemic homozygous LepR$^{db/db}$ male mice that are destined for diabetes, proinsulin disulfide-linked complexes (*Figure 5C upper left panel*) were increased slightly above the level observed in heterozygous LepR$^{db/+}$ males (that do not progress to diabetes), and islet insulin levels were similarly increased (*lower left panels*). With a random blood glucose level in the mid 200's (mg/dL), islet proinsulin was notably increased (*Figure 5C upper second panel*) and comprised >90% of disulfide-linked proinsulin complexes (*upper first panel*) whereas simultaneously, islet insulin levels were dramatically decreased (*lower left panels*). With a random blood glucose level >500 mg/dL, islet proinsulin and insulin levels were both severely diminished (*Figure 5C left panels*).

In female LepR$^{db/db}$ mice with a random blood glucose ranging from 156 to 294 mg/dL, insulin levels declined compared to the LepR$^{db/+}$ control (*Figure 5C lower right panels*). However, for females in this glycemic range, as in males, islet proinsulin was increased (*Figure 5 upper right panel*) and once again, >90% was entangled in disulfide-linked proinsulin complexes (*Figure 5 upper third panel*). Additionally, with a random blood glucose of 455 mg/dL, both proinsulin and insulin were severely diminished (*Figure 5C last two sets of panels*) indicating islet decompensation.

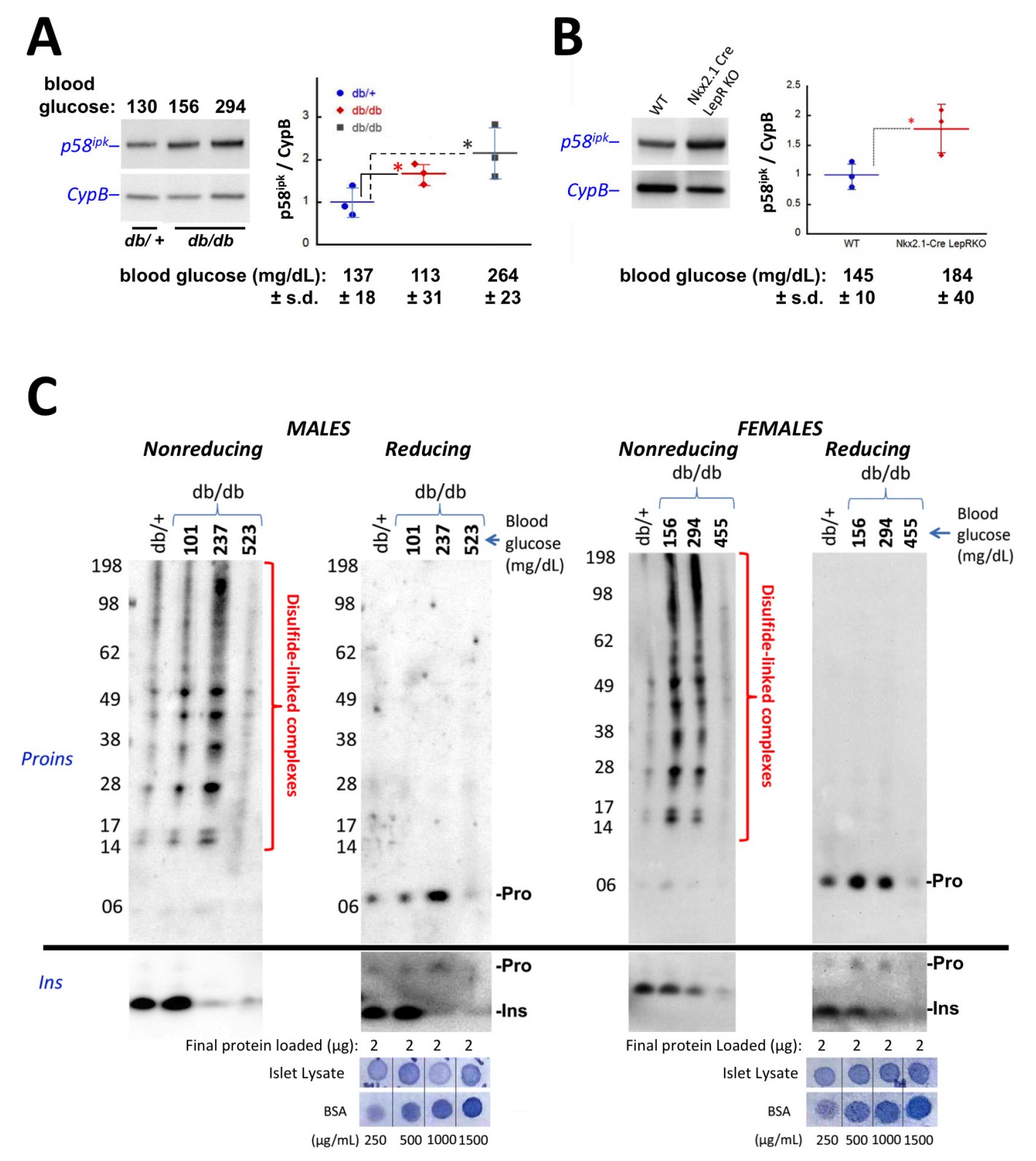

**Figure 5.** Accumulation of improperly folded proinsulin and detection of ER stress response are early events in the development of diabetes in leptin receptor-deficient mice. (A) Islet lysates from LepR$^{db/+}$ heterozygote (control) or LepR$^{db/db}$ mice at different stages of diabetes progression (*random blood glucose values shown above*) were immunblotted for p58$^{ipk}$ and cyclophilin B (CypB, loading control); the graph at right shows the quantitation of the p58$^{ipk}$ / CypB ratio from three independent experiments (mean ± s.d., each point a different animal; asterisk, p=0.05 by Mann Whitney U test and *Figure 5 continued on next page*

*Figure 5 continued*

p<0.05 by t-test). (**B**) Islet lysates from WT and Nkx2.1-Cre-mediated LepR-KO mice were immunblotted for p58[ipk] and CypB as in panel A; the graph at right shows quantitation of the p58[ipk] / CypB ratio from three independent experiments (mean ± s.d., each point a different animal; p=0.05 by Mann Whitney U test and p<0.05 by t-test). (**C**) Islet lysates from LepR[db/+] heterozygote (control) or LepR[db/db] mice at different stages of diabetes progression (*random blood glucose values shown above; blood glucose in LepR[db/+]male was 138 mg/dL and in female was 117 mg/dL*) were analyzed by nonreducing or reducing SDS-PAGE and immunoblotting with mAb anti-proinsulin (CCI-17, *above black line*; molecular mass markers are noted) or guinea pig anti-insulin (*below black line*). Islet protein content from each mouse was measured (*Bramhall et al., 1969*) relative to known BSA standards shown at bottom; 2 μg islet protein was analyzed for each sample. Left sets of gels are from males; right sets of gels are from females, as indicated. In LepR[db/db] mice, intra-islet abundance of misfolded disulfide-linked proinsulin complexes reached maximum with random blood glucoses ranging from 150 to 300 mg/dL but this was accompanied by a decline in intra-islet mature insulin levels. Overtly diabetic animals (random blood glucose >450) exhibited low insulin levels and ultimately exhibited low proinsulin levels as well.

DOI: https://doi.org/10.7554/eLife.44532.012

The following source data and figure supplement are available for figure 5:

**Source data 1.**
DOI: https://doi.org/10.7554/eLife.44532.014
**Source data 2.**
DOI: https://doi.org/10.7554/eLife.44532.015
**Figure supplement 1.** Intracellular proinsulin distribution in the LepR[db/db] mouse.
DOI: https://doi.org/10.7554/eLife.44532.013

During this progression to diabetes, intracellular proinsulin distribution that initially displayed a prominent juxtanuclear pattern with good co-localization to the Golgi region (GM130 Golgi marker) tended to shift towards a staining pattern that was more spread in the cytoplasm, with increased co-localization with the ER marker, calnexin (*Figure 5—figure supplement 1*); although even in the fully diabetic state there appeared to be increased juxtanuclear (Golgi-like) proinsulin distribution after overnight fasting (*Figure 5—figure supplement 1*).

## Mechanism of proinsulin intermolecular disulfide complex formation

Cys(A20) and Cys(B19) are two of the three most reactive thiols of proinsulin, and they initiate covalent association of B- and A-domains that are required for proinsulin export from the ER (*Haataja et al., 2016*). Nevertheless, in the redox environment of the pancreatic ß-cell ER, a human mutant proinsulin that retains only two cysteines, named 'keep-B19/A20' cannot undergo efficient intramolecular oxidation (*Haataja et al., 2016*). Reactive free thiol availability (*Figure 1A*) is a key to proinsulin participation in intermolecular disulfide-linked complex formation. Indeed, each of the myc-tagged human proinsulins keep-B7/A7 and keep-A6/A11, in addition to keep-B19/A20, exhibited free thiol availability as determined by alkylation with AMS (*Figure 6—figure supplement 1*). We expressed each of these recombinant proinsulin mutants in the INS-832/13 ß-cell line (which already expresses endogenous proinsulin). Because these two-cysteine mutant human proinsulins cannot be exported from the ER and are excellent ERAD substrates in pancreatic ß-cells (*Haataja et al., 2016*), these molecules were largely undetected by immunoblotting. However, in ß-cells treated with proteasome inhibitor (MG132 for 7 hr), myc-tagged proinsulin monomers were detected in all cases (*Figure 6A*). Remarkably, not only were these constructs capable of forming disulfide-linked proinsulin homodimers, but the keep-B19/A20 construct in particular recapitulated the situation in which the majority of molecules migrated as disulfide-linked proinsulin complexes (*Figure 6A*). With or without MG132, this behavior was also observed upon expression in 293 T cells that lack endogenous proinsulin (*Figure 6B*), and these complexes appeared quite similar to those observed in human islets (*Figure 3A,B*).

To determine if Cys(B19)/Cys(A20) are both necessary and sufficient for generating the full ladder of disulfide-linked proinsulin complexes, we compared the three different human mutant proinsulins noted above that each retain only one set of potential disulfide partners, and a human proinsulin mutant named 'lose-B19/A20' in which Cys(B19) and Cys(A20) are mutated whereas positions B7, A7, A11 and A20 remain as cysteines. Once again, of the mutant human proinsulins that retain only one set of potential disulfide partners, only keep-B19/A20 could efficiently recreate the entire ladder of disulfide-linked complexes similar to that seen for wildtype proinsulin, irrespective of the presence or absence of an engineered myc epitope tag in the C-peptide (*Figure 6C left panel*). However,

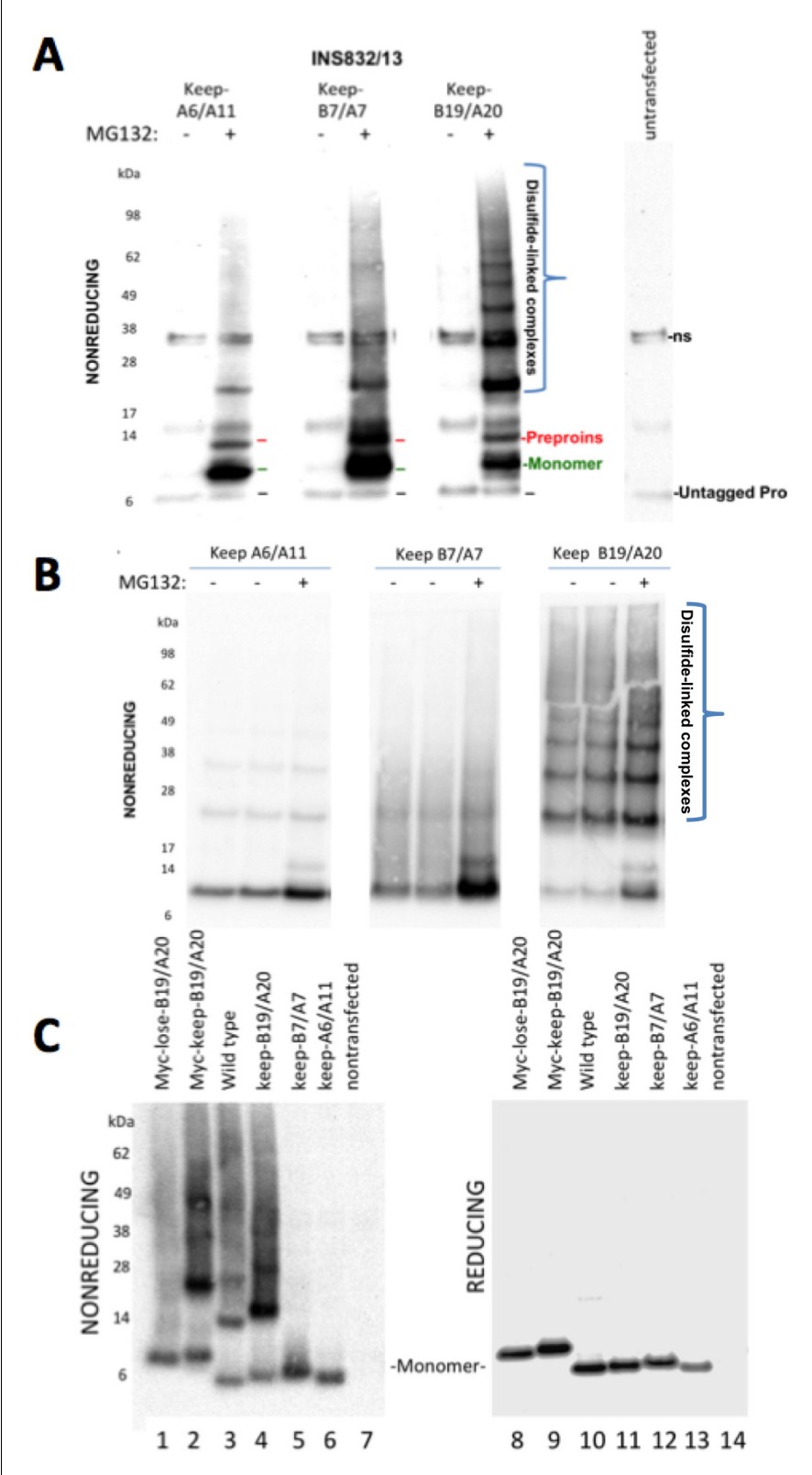

**Figure 6.** Proinsulin Cys residues that contribute to covalent complex formation. (**A**) INS832/13 were transfected to express myc-tagged recombinant human proinsulin 'keep-A6/A11', 'keep-B7/A7', or 'keep-B19/A20' constructs. At 48 hr post-transfection, cells were treated with vehicle alone (–) or MG132 (10 μg/mL) for 7 hr before cell lysis and analysis by nonreducing SDS-PAGE and immunoblotting with mAb directed against a sequence (PLALEGSLQKRGIV) spanning the junction of the proinsulin C-peptide and insulin A-chain. A control of mock-transfected cells (*far right lane*) is shown

*Figure 6 continued on next page*

*Figure 6 continued*

for comparison; 'ns'=nonspecific band. The positions of molecular mass markers are noted. (**B**) 293 T cells were transfected to express the same constructs and treated and analzyed as in panel A. The positions of molecular mass markers are noted. (**C**) 293 T cells were transfected to express the same constructs as in panel A, or myc-tagged 'lose-B19/A20', myc-tagged 'keep-B19/A20', and untagged wild-type human proinsulin. Cell lysates were resolved by SDS-PAGE under nonreducing and reducing conditions and analyzed as in panel A; molecular mass markers are noted.

DOI: https://doi.org/10.7554/eLife.44532.016

The following figure supplement is available for figure 6:

**Figure supplement 1.** Free thiols in recombinant proinsulin mutants.

DOI: https://doi.org/10.7554/eLife.44532.017

lose-B19/A20 did not efficiently recreate the ladder of disulfide-linked complexes despite entrapment in the ER in a proinsulin construct bearing four Cys residues. These data indicate that the Cys (B19)/Cys(A20) pair is both necessary and sufficient for efficient propagation of disulfide-linked complexes of improperly-folded proinsulin.

In these experiments, we were struck by the observations 1) that all 2-Cys proinsulin mutants have the capability to at least weakly form a covalent dimer, which is essential to the further propagation of these improperly folded complexes into trimers, tetramers, and higher order complexes (*Figure 2B*), and 2) a small fraction of proinsulin dimers appeared to remain even after reducing SDS-PAGE (*Figure 4D*; *Figure 1—figure supplement 1C*). To understand these behaviors, we expressed a series of mutant human proinsulins that retain only a single cysteine (and thus cannot make more than one intermolecular disulfide bond). Although five of the six Cys residues could make a homotypic covalent association, Cys(A7) preferred not to make the disulfide bond (*Figure 7A left panel*; quantified in *Figure 7B*). More significantly, Cys(B19) not only covalently homodimerized with exuberance (*Figure 7A left panel*) but this Cys(B19)-Cys(B19) bond could not be broken at room temperature in SDS-gel sample buffer even in the presence of 200 mM dithiothreitol (*Figure 7A right panel*). These data indicate that a Cys(B19)-Cys(B19) disulfide bond between proinsulin monomers predisposes to strongly-associated covalent complexes that are automatically misfolded by omitting the crucial intramolecular Cys(B19)-Cys(A20) disulfide bond, which leaves the critical reactive Cys(A20) residue unpaired and available for further disulfide infidelity. Such an initiating event can thus form a nidus for assembly of larger misfolded disulfide-linked proinsulin complexes (*Figure 7—figure supplement 1B*).

## Discussion

It is recognized from human genome-wide association studies and animal models that progression from insulin resistance to pancreatic ß-cell dysfunction is a linchpin in the development of T2D. Pancreatic ß-cell ER stress is one of the most frequently described components of T2D in humans and animal models (*Back and Kaufman, 2012*; *Berry et al., 2018*; *Cnop et al., 2017*; *Laybutt et al., 2007*; *Marchetti et al., 2007*; *Rabhi et al., 2014*; *Yang et al., 2016*). While protein misfolding caused by mutations (in so-called conformational diseases) is one recognized cause of ER stress, the proximal trigger of beta cell ER stress early in the progression of T2D is unknown, but has been the subject of much speculation. It is indisputable that ß-cell ER stress can be triggered by proinsulin misfolding in the setting of *INS* gene coding sequence mutations (*Liu et al., 2018*) but there are also strong reasons to think that even in the absence of *INS* gene mutations, proinsulin misfolding could be an early feature in the progression of T2D (*Arunagiri et al., 2018*; *Riahi et al., 2018*; *Scheuner and Kaufman, 2008*).

In the present study, we have exploited several independent lines of experimentation to identify a significantly increased population of improperly folded proinsulin in the ER of prediabetic and diabetic ß-cells. We establish that proinsulin misfolding includes the inability to successfully complete its three internal disulfide bonds, with a subfraction of proinsulin molecules in the ER bearing unpaired cysteine residues. This population of improperly folded proinsulin molecules enters into aberrant disulfide-linked partnerships with other proinsulin molecules in the ER, resulting in misfolded proinsulin complexes that are for the first time identified in the islets of human beings.

There is strong reason to believe that proinsulin disulfide-linked complex formation provides a status report on the health of the ß-cell ER folding environment. Many studies point to the idea that

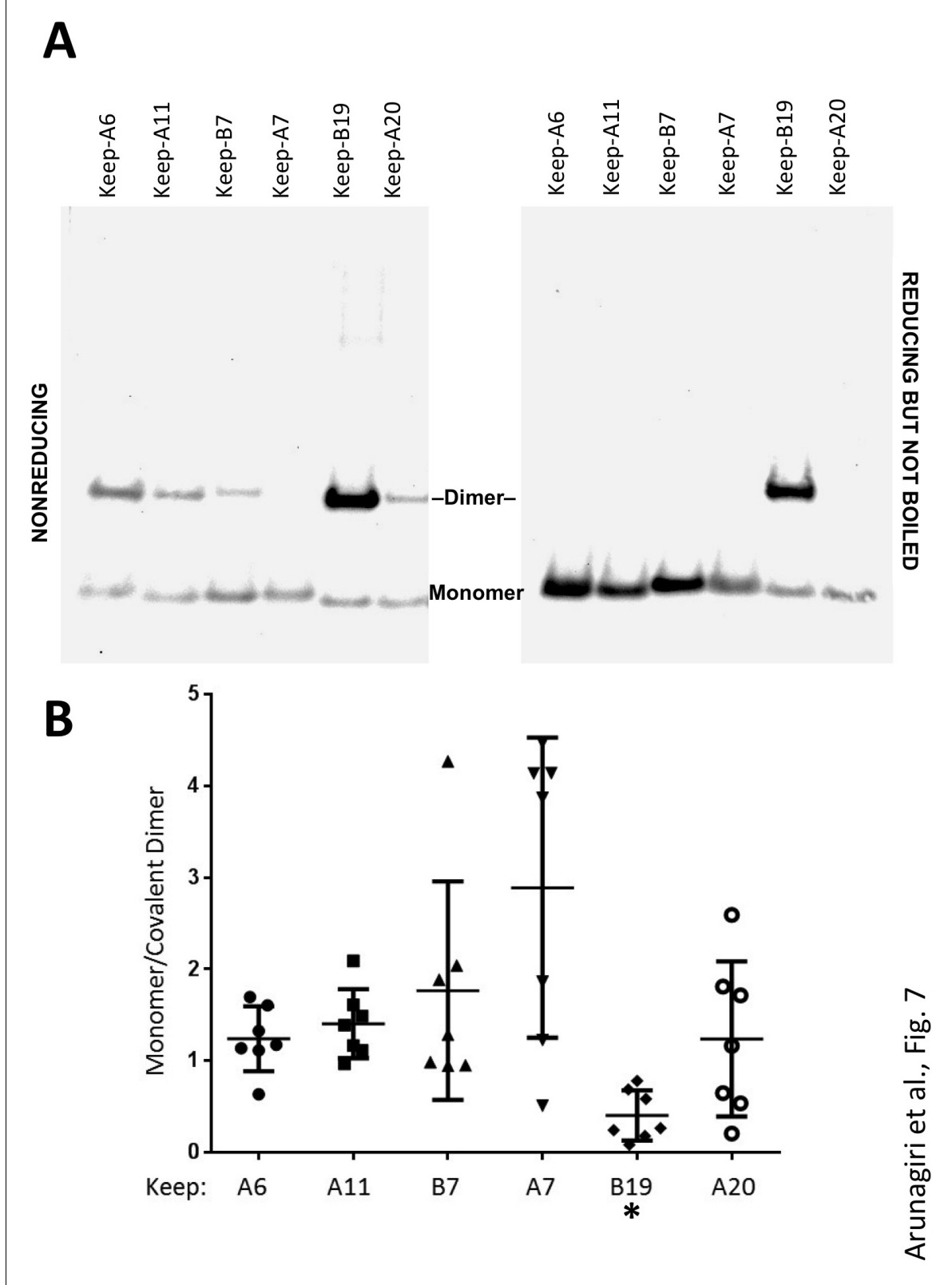

**Figure 7.** Proinsulin intermolecular disulfide crosslinking is promoted by Cys(B19). (**A**) 293 T cells were transfected to individually express six distinct human proinsulin mutants bearing only one cysteine. At 28 hr post-transfection, cell lysates were analyzed either by nonreducing SDS-PAGE (*left*) or incubated in SDS-gel sample buffer plus 200 mM DTT at room temperature for 10 min prior to SDS-PAGE. The gels were identically electrotransferred to nitrocellulose, and immunoblotted for proinsulin as in *Figure 6*. (**B**) Seven independent experiments like that shown in the left panel of *Figure 7A*

*Figure 7 continued on next page*

*Figure 7 continued*

were quantified for monomer to covalent dimer ratio: keep-A7 was mostly monomeric whereas keep-B19 predominantly formed covalent dimers (asterisk signifies p<0.05 when compared to each keep mutant except keep-A20).

DOI: https://doi.org/10.7554/eLife.44532.018

The following source data and figure supplement are available for figure 7:

**Source data 1.**

DOI: https://doi.org/10.7554/eLife.44532.020

**Figure supplement 1.** Native and non-native proinsulin disulfide pairing.

DOI: https://doi.org/10.7554/eLife.44532.019

the activity of ER stress (UPR) sensor proteins are required to actively maintain a proper ER chaperone and oxidoreductase environment for optimal proinsulin folding (*Harding et al., 2012*; *Hassler et al., 2015*; *Sowers et al., 2018*; *Tsuchiya et al., 2018*). Recent work indicates that protein disulfide isomerase is one of the ER luminal factors that regulates the balance of proinsulin disulfide-linked complexes and monomers (*Jang et al., 2019*). Additionally, BiP plays a central role in ER stress sensing (*Amin-Wetzel et al., 2017*). We show here that even a modest decrease of BiP levels results in improperly folded proinsulin rapidly accumulating in disulfide-linked complexes (*Figure 3—figure supplement 1B*), supporting that proinsulin folding is highly sensitive to changes in the ER folding environment, which can include excessive proinsulin biosynthesis, altered chaperone and/or oxidoreductase expression, and changes in the rate of clearance of misfolded proinsulin molecules (*Arunagiri et al., 2018*; *Xu et al., 2018*). We believe that these and other ER luminal factors are needed to enhance the efficiency of formation of the proinsulin intramolecular Cys(B19)-Cys(A20) disulfide bond. Not only is this internal disulfide critical to additional native proinsulin disulfide pairing (*Haataja et al., 2016*) (*Figure 7—figure supplement 1A*), but the availability of free Cys(B19) and Cys (A20) can recapitulate the entire ladder of improperly disulfide linked proinsulin complexes. Several pathways of disulfide propagation are possible, highlighted in *Figure 7—figure supplement 1B*. However, those complexes bearing a homotypic Cys(B19)-Cys(B19) covalent bond are likely to be the most difficult to successfully isomerize, as this bond cannot be broken in vitro even in the presence of SDS plus 200 mM DTT (*Figure 7A*).

A specific goal of T2D research is to better understand the natural history of ß-cell failure (*Halban et al., 2014*). What has recently emerged using the LepR$^{db/db}$ mouse model, is that beginning around 4–5 months of life, after hyperglycemia has appeared, ß-cells appear to be in the early stages of dedifferentiation (*Ishida et al., 2017*), and with a similar time course, ß-cell apoptosis may occur in parallel (*Wu et al., 2015*). However, clearly there are important islet changes that occur considerably before these events (*Do et al., 2016*) including ß-cell ER stress response activation (*Figure 5A,B*) (*Herbert and Laybutt, 2016*). *Figure 8* presents a schematic describing our hypothesis regarding how proinsulin misfolding fits into the paradigm of what is known about T2D progression in the LepR$^{db/db}$ model. Our work highlights that throughout the first, compensated stage, improperly folded proinsulin is increased but islet insulin content is still robust (*Figures 4D* and *5C*). In the second, decompensating stage, islet proinsulin levels are actually further increased, but much of this is improperly folded proinsulin in the ER, and at this stage, islet insulin levels begin to drop (*Figure 5*). It is at a still higher level of hyperglycemia that both proinsulin and insulin steady state levels are low (*Figure 8*), which correlates with the time when ß-cell dedifferentiation and ß-cell death have been reported. It should be noted that even at this time, the low-level proinsulin protein that is still expressed is recovered >90% in aberrant disulfide linked complexes, and even partial restoration of secretory pathway function requires ß-cell rest (*Alarcon et al., 2016*).

We suspect that throughout the progression of ß-cell stress in T2D, quality control of proinsulin anterograde export from the ER is nevertheless maintained. We note that we have been unable to detect secreted proinsulin that can be alkylated with AMS, or that is recovered in disulfide-linked complexes, even from islets or cells in which proinsulin has accumulated intracellularly in disulfide-linked complexes at massive levels. However, experiments such as these will need to be rigorously tested with islets from human T2D subjects, as these patients are known to have abnormally high levels of immunoreactive proinsulin in the circulation (*Porte and Kahn, 2001*).

In conclusion, our data establish the presence of a previously underappreciated population of aberrant proinsulin complexes that accumulates in prediabetic conditions and persists until ß-cell

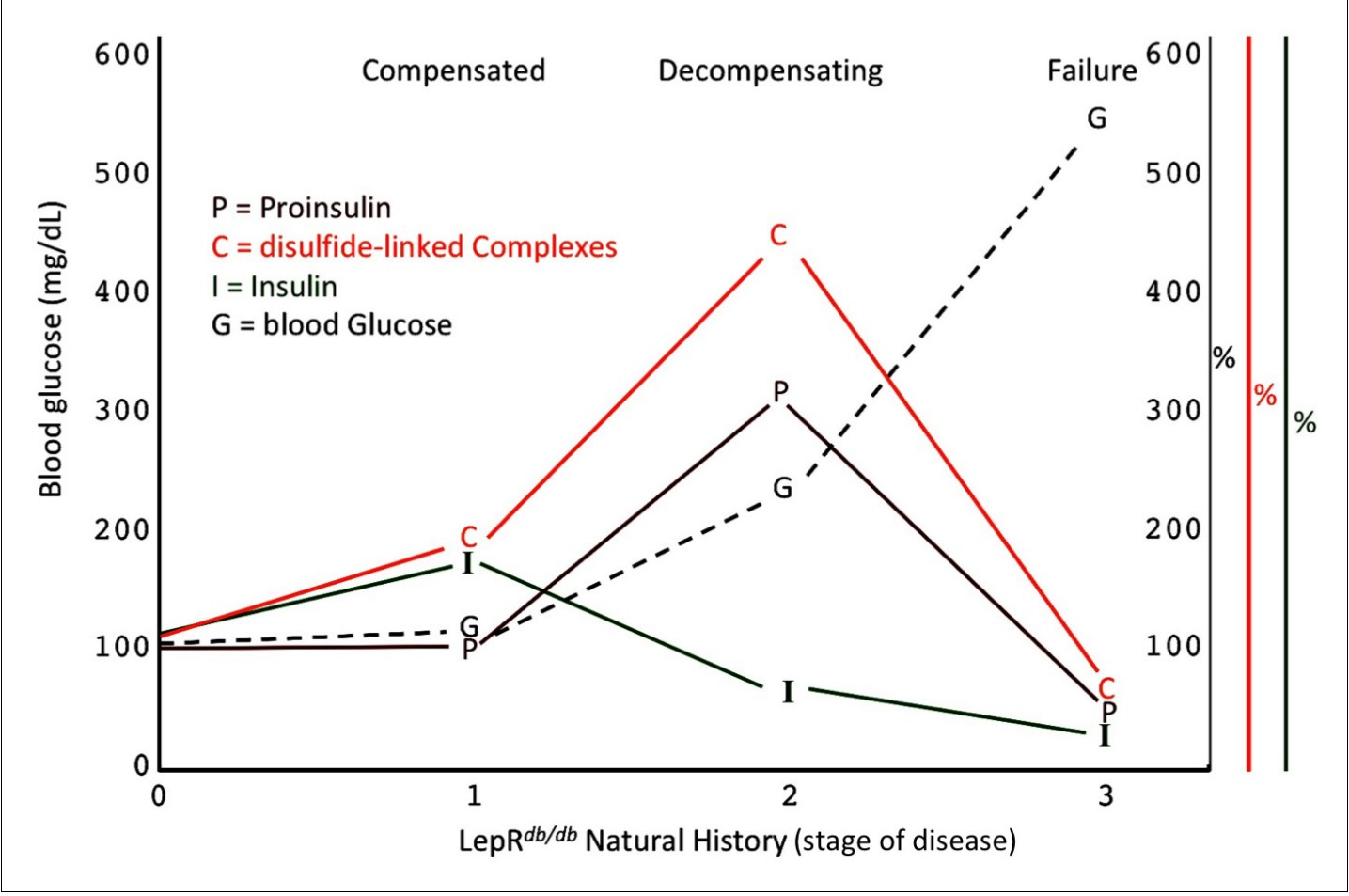

**Figure 8.** Islet dysfunction during the natural history of diabetes in the LepR$^{db/db}$ mouse, as a model. A schematic is shown, indicating progression of early islet dysfunction during the natural history of diabetes in the LepR$^{db/db}$ mouse. In the first stage of postnatal life, random blood glucose is in the normal range and insulin content in islets is actually slightly greater than that seen in the control condition. Total proinsulin levels are not diminished, but there is a slight increase in proinsulin disulfide-linked complexes. At the second stage, random hyperglycemia (150–300 mg/dL) is observed, accompanied by islet insulin levels that are less than that seen in the control condition. However, total proinsulin levels are notably increased, accompanied by an increase in proinsulin disulfide-linked complexes. At a third stage is a worsening of random hyperglycemia (>450 mg/dL), accompanied by low islet insulin and low islet proinsulin, which has been attributed to ß-cell dedifferentiation, or ß-cell death, or could potentially represent a combination of both.

DOI: https://doi.org/10.7554/eLife.44532.021

failure ensues. We note that the proinsulin complexes defined by nonreducing SDS-PAGE highlight a core of covalently-associated proteins; further work will be needed to explore additional protein partners including those that may be noncovalently associated. Nevertheless, the data presented herein establish disulfide-linked complexes of proinsulin as one of the earliest tissue biomarkers indicating ß-cell secretory pathway dysfunction, which is associated with ER stress and ultimate insulin deficiency that occurs in the natural progression of T2D.

## Materials and methods

### Chemicals and reagents
AMS, DTT, N-ethylmaleimide, PERK inhibitor (GSK2656157), MG132, Cycloheximide, and all other chemicals were purchased from Sigma-Aldrich or ThermoFisher Scientific. SDS-PAGE 4–12% Bis-Tris or 4–20% Tris-Glycine NuPage gels were obtained from ThermoFisher.

## Antibodies

Antibodies used in the present work are as follows:

1. mouse mAb 20G11 directed against an epitope in the region of EAEDLQVGQVELGG of human C peptide was obtained at the Scripps antibody production core facility. This mAb does not significantly cross-react with rodent proinsulin.
2. mouse mAb GS-9A8 directed against an epitope in the region of YTPKTRREAEDL of human proinsulin (which spans the B-C cleavage junction and cross-reacts with rodent proinsulin in formaldehyde-fixed tissue) was obtained from the DSHB at the University of Iowa.
3. mouse mAb mAb CCI-17 directed an epitope in the region of the RQKRGIVEQ sequence of rat proinsulin (which spans the C-A cleavage junction) was obtained from ALPCO. This mAb does not significantly cross-react with human proinsulin.
4. mouse mAb directed against an epitope in the region of PLALEGSLQKRGIV sequence of human proinsulin (which spans the C-A cleavage junction) was obtained from Abmart. This mAb shows partial cross-reactivity with rodent proinsulin.
5. guinea pig anti-insulin was used both for immunofluorescence (DAKO) and Western blotting (Covance). Guinea pig polyclonal anti-insulin cross reacts with proinsulin and conversion intermediates, but it preferentially reacts with mature insulin, of multiple species.
6. mouse mAb anti-KDEL was from Enzo Life Sciences.
7. rabbit polyclonal anti-calnexin was as previously described (*Kim et al., 1996*).
8. mouse mAb anti-GM130 was obtained from BD Biosciences.
9. rabbit polyclonal anti-cyclophilin B was obtained from ThermoFisher.
10. rabbit mAb anti-p58[ipk] was obtained from Cell Signaling Technologies.

## Mouse models

The mutant LepR[db] allele was carried in the C57BLKS/J background; heterozygotes were cross-bred to generate LepR[db/db] homozygotes. Wild-type and *Akita* mutant mice were maintained in the C57BL/6J background. All of the preceding breeder mice were obtained from JAX. $Ins1^{-/-}$, $Ins2^{+/-}$ mice generated in a C57BL/6 background were intercrossed as previously described (*Duvillie et al., 1997*; *Duvillié et al., 2002*). In this study we restricted our analyses to males because *Akita* diabetic males develop progressive hyperglycemia to which the females are resistant (*Yoshioka et al., 1997*). LepRNkx2.1 KO mice were identical to those previously described (*Ring and Zeltser, 2010*).

## Mouse pancreatic islet isolation

Mice were euthanized by $CO_2$ narcosis as per an approved institutional animal protocol. The pancreas was rapidly excised, minced in ice-cold PBS, and digested in 4 mL of Collagenase P (Roche) 1.5 mg/mL in Hank's Balanced Salt Solution containing calcium and magnesium, in a shaking water bath for 30 min at 37 ˚C. The digestion was terminated in 40 mL ice-cold PBS (-Ca/-Mg) and the digested tissue washed twice in this buffer. The sedimented tissue digest was then overlaid with 3 mL Histopaque-1077 and further overlayed with 6 mL ice cold PBS (-Ca/-Mg) before centrifugation (3000 rpm for 20 min at 10 ˚C with no brake) and transfer of the mid-layer of the Histopaque gradient to a 15 ml tube for two further washes in ice-cold PBS. Finally, the sedimented tissue was transferred to a petri dish in ice-cold RPMI- 1640 buffer and islets hand-picked to purity. In experiments involving drug or compound treatment, the islets were finally recovered overnight in RPMI-1640 supplemented with 10% fetal bovine serum (FBS), 10 mM HEPES pH 7.35, 1 mM sodium pyruvate and 0.05 mM beta-mercaptoethanol, in a humidified 5% CO2 incubator at 37 ˚C. For some experiments, islets were snap frozen in liquid nitrogen and stored frozen prior to analysis.

## Human pancreatic islets

Human islets were obtained either from Prodo Labs, or from the NIDDK-funded Integrated Islet Distribution Program (IIDP; NIH UC4-DK098085) and maintained in a humidified 5% CO2 incubator at 37˚C. Human islets were cultured ex vivo for up to 96 hr in Prodo PIM(R) islet-specific tissue culture medium supplemented with 10% FBS and Prodo PIM(G) glutamine/glutathione supplement, plus penicillin/streptomycin.

## Cell and islet culture

Rat pancreatic beta cell lines INS1E and INS-832/13 were cultured in RPMI-1640 medium supplemented with 10% FBS, 10 mM HEPES pH 7.35, 1 mM sodium pyruvate, penicillin/streptomycin and 0.05 mM beta-mercaptoethanol. HEK293T human cells lines were cultured in DMEM supplemented with 10% FBS and penicillin/streptomycin.

## AMS-mediated alkylation of proinsulin

INS1E cell or mouse or human islet lysate, each diluted in reaction buffer (50 mM Tris pH 7.4, 1% SDS final concentrations) were heated to 95 °C for 5 min, cooled to room temperature and then incubated further in the same buffer containing 4-Acetamido-4'-Maleimidylstilbene-2,2'-Disulfonic Acid, Disodium Salt (AMS, ThermoFisher) for 1 hr at 37°C. In one experiment, 2 mM DTT was added during the initial boiling step prior to aklylation. No differences in alkylation were observed at AMS doses ranging from 6 mM to 20 mM. Non-alkylated controls underwent all the same incubations, in the same buffers, in parallel. The INS1E cells bathing media were also similarly tested for AMS-reactive proinsulin species. AMS-treated and untreated controls were analyzed by nonreducing or reducing SDS-PAGE and immunoblotting as described below.

## Proinsulin mutagenesis, and plasmids

The generation of myc-tagged keep-B7/A7, keep-B19/A20, and keep-A6/A11 were described previously (Haataja et al., 2016). These plasmids were used as templates for further mutagenesis to create six single-cysteine myc-tagged proinsulin mutants using the QuikChange site-directed mutagenesis kit (Agilent). All resulting plasmids encoding corresponding proinsulin mutations were confirmed by direct DNA sequencing. The expression plasmid encoding hPro-CpepMyc has been previously described (Haataja et al., 2013).

## Transfection of cells

293 T cells, INS1E cells, and INS-832/13 cells at 70–80% confluency were transiently transfected using Lipofectamine 2000 (ThermoFisher) as per the manufacturer's instructions. A medium change was performed 5 hr post-transfection and the cells were lysed at 36 or 48 hr.

## Lysis of cells or islets for SDS-PAGE and electrotransfer

After removal of media, cells were washed once with ice-cold PBS and lysed in RIPA buffer (10 mM Tris pH 7.4, 150 mM NaCl, 0.1% SDS, 1% NP40, 2 mM EDTA) plus protease inhibitor/phosphatase inhibitor cocktail (Sigma-Aldrich) or directly in Laemmli gel sample buffer. Lysates in RIPA buffer were immediately spun at 10,000 rpm for 10 min at 4 °C and the supernatants analyzed further or stored at −80°C. Islets that had been quick frozen were placed on ice and RIPA buffer containing protease inhibitor cocktail and phosphatase inhibitor was added. Lysis was carried out by gently pipetting or by syringe through a 30G needle. Total protein concentration in the lysate was determined by BCA or Bramhall assay, and 5–10 µg of samples prepared in SDS sample were resolved by SDS-PAGE in 4–12% Bis-Tris NuPAGE gels (Invitrogen) at 200 V for 30 min. Nonreducing gels were incubated in a solution containing 25 mM dithiothreitol (DTT) for 10 min at room temperature prior to electrotransfer.

## Two-dimensional (2-D) gel electrophoresis

A lane excised from the nonreducing slab gel loaded with lysate from INS1E cells treated with PERK inhibitor, was incubated for 15 min at room temperature in Tris-Glycine pH 6.8 plus 20 mM DTT. The lane was then laid horizontally on top of a 10% SDS-polyacrylamide resolving gel and sealed in place in stacking gel. A single-tooth comb was introduced at one end of the stacking gel to introduce a one-dimensional reduced sample of proinsulin containing cell lysate (which runs as proinsulin monomer). The gel was run at 200 V for 45 mins in Tris-Glycine buffer pH 8.8 followed by electrotransfer.

## Immunoblotting

Transfer membranes were rinsed once in TBST (15 mM Tris, 150 mM NaCl, 0.1% Tween-20), blocked using TST plus 5% BSA for 1 hr at room temperature, and then washed four times (5 min each) in TBST. Primary antibody and secondary antibodies were diluted in TBST and each incubation was 1

hr at room temperature followed by four washes in TBST. On occasion, primary antibodies were incubated overnight at 4 °C. Secondary antibodies were all HRP-conjugates. Development of immunoblots used enhanced chemiluminescence (Immobilon, Millipore, or SuperSignal West Pico PLUS, Thermo Fisher Scientific) with images captured using a Fotodyne gel imager.

### Immunohistochemistry/immunocytochemistry

Pancreatic paraffin sections were de-paraffinized with Citrisolv (Fisher Scientific) for 5 min at room temperature, re-hydrated in a series of 8 min ethanol incubations: 100%, 95%, 70%, and distilled water. Antigen retrieval was carried out using Retrieve-ALL 1 Universal pH 8 (BioLegend) and slides were heated in a microwave, cooled for 30 min at room temperature, and incubated once with PBS for 5 min. Blocking was performed in 150 µL blocking buffer (3% BSA prepared in TBS and 0.2% Triton X-100) per section for 2 hr at room temperature. Blocking buffer was removed and 150 µL per section of primary antibody appropriately diluted in TBS plus 3% BSA and 0.2% Tween-20 was incubated overnight at 4 °C. The primary antibody was removed and the slide washed twice with TBS/0.1% Tween-20. Then, 150 µL secondary antibody (1:500 dilution, prepared in antibody buffer) per section was incubated for 1 hr at room temperature. Slides were washed thrice with TBS/0.1% Tween-20, mounted with a drop of prolong gold anti-fade reagent with DAPI (ThermoFisher) and a cover slip affixed.

For immunocytochemistry, INS1E cells were grown in an 8-well Millicell EZ SLIDE (Millipore-SIGMA). Cells were allowed to reach 60–80% confluency before addition of PERK inhibitor (PERKi) or vehicle. After 18 hr of PERKi treatment, the medium was removed and the cells were fixed in 3.7% formaldehyde in PBS pH 7.4, for 20 min at room temperature, rinsed once with PBS, and permeabilized with 0.4% Triton X-100 in TBS for 20 min at room temperature. The cells were washed thrice with TBS and then incubated in blocking buffer as described above. Thereafter samples were incubated overnight at 4°C with 100 µL of appropriately diluted primary antibody in 3% BSA/TBS/0.2% Tween. The cells received four 15 min washes in TBS, and then incubated with secondary antibody (1:500 dilution) for 1 hr at room temperature. The cells were then washed thrice with TBS and finally mounted with a drop of Prolong Gold anti-fade reagent containing DAPI (ThermoFisher) and a cover slip affixed and the slide incubated in the dark for 24 hr at room temperature. A similar immunocytochemistry protocol was followed for INS-832/13 cells after SubAB treatment (4 hr).

## Acknowledgements

This work was supported by NIH R24 DK110973 (to PA, P I-A, and RJK), R01 DK48280 (to PA), R01 DK111174 (to PA, ML, and BT), and a research grant from the JDRF (2-SRA-2018–539-A-B). We also acknowledge support of the Protein Folding Diseases Initiative of the University of Michigan, and the NIH-funded Michigan Diabetes Research Center (P30-DK020572). We acknowledge the labs of WE Balch (Scripps Institute, La Jolla CA USA) for development of the mAb 20G11 antibody, and J Wahren (Karolinska Instituet, Stockholm Sweden) for purified human C-peptide as a competitor blocking peptide.

## Additional information

### Funding

| Funder | Grant reference number | Author |
| --- | --- | --- |
| National Institute of Diabetes and Digestive and Kidney Diseases | R01DK111174 | Billy Tsai<br>Ming Liu<br>Peter Arvan |
| National Institute of Diabetes and Digestive and Kidney Diseases | R24DK110973 | Pamela Itkin-Ansari<br>Randal J Kaufman<br>Peter Arvan |
| National Institute of Diabetes and Digestive and Kidney Diseases | R01 DK48280 | Peter Arvan |

| Juvenile Diabetes Research Foundation International | 2-SRA-2018-539-A-B | Peter Arvan |
|---|---|---|

The funders had no role in study design, data collection and interpretation, or the decision to submit the work for publication.

## Author contributions

Anoop Arunagiri, Leena Haataja, Validation, Investigation, Writing—original draft, Writing—review and editing; Anita Pottekat, Fawnnie Pamenan, Validation, Investigation, Writing—review and editing; Soohyun Kim, Writing—review and editing; Lori M Zeltser, Adrienne W Paton, James C Paton, Billy Tsai, Resources, Writing—review and editing; Pamela Itkin-Ansari, Randal J Kaufman, Conceptualization, Supervision, Funding acquisition, Methodology, Project administration, Writing—review and editing; Ming Liu, Conceptualization, Funding acquisition, Methodology, Project administration, Writing—review and editing; Peter Arvan, Conceptualization, Supervision, Funding acquisition, Methodology, Writing—original draft, Project administration, Writing—review and editing

## Author ORCIDs

Anoop Arunagiri (iD) http://orcid.org/0000-0001-9839-1860
Billy Tsai (iD) http://orcid.org/0000-0003-2859-1415
Randal J Kaufman (iD) http://orcid.org/0000-0003-4277-316X
Peter Arvan (iD) https://orcid.org/0000-0002-4007-8799

## Ethics

Animal experimentation: This study was performed in strict accordance with the recommendations in the Guide for the Care and Use of Laboratory Animals of the National Institutes of Health. All of the animals were handled and euthanized according to approved institutional animal care and use committee (IACUC) of the University of Michigan (protocol # PRO00008062).

## Decision letter and Author response

Decision letter https://doi.org/10.7554/eLife.44532.026
Author response https://doi.org/10.7554/eLife.44532.027

## Additional files

### Supplementary files

• Transparent reporting form
DOI: https://doi.org/10.7554/eLife.44532.022

### Data availability

All data generated or analysed during this study are included in the manuscript and supporting files.

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
