## [Decision Letter]

Thank you for submitting your article "Proinsulin misfolding is an early event in the progression to Type 2 Diabetes" for consideration by *eLife*. Your article has been reviewed by three peer reviewers, and the evaluation has been overseen by Reid Gilmore as the Reviewing Editor and Vivek Malhotra as the Senior Editor. The reviewers have opted to remain anonymous.

The reviewers have discussed the reviews with one another and the Reviewing Editor has drafted this decision to help you prepare a revised submission.

Arunagiri and colleagues have investigated the formation of misfolded proinsulin complexes during the development and progression of Type 2 Diabetes (T2D). They found that native proinsulin forms intermolecular disulfide bonds in murine and human islets and show that this process is enhanced when the ER folding machinery is disrupted. The authors have determined that aggregate formation is linked to the generation of non-native disulfides involving B19 and A20. Importantly, the authors show in two models of diabetes that disulfide-linked proinsulin complexes accumulate during early stages of diabetes prior to development of hyperglycemia suggesting that formation of misfolded proinsulin complexes is an early event in the course of diabetes that might have a role in the β-cell dysfunction. The findings are novel and interesting; however, additional experiments are required to clarify the impact of these complexes on β-cell stress and function.

Essential revisions:

1) The majority of data is based on non-reducing SDS-PAGE immunoblots. The authors performed multiple experiments to demonstrate the specificity of the assay. The authors should reconsider their interpretation of the experiments shown in Figure 2C. The NEM treatment prior to cell lysis (lane 2) seems to stabilize very large aggregates (>pentamers). Cell lysis without blocking free thiols favors formation of the proinsulin ladders (dimers to pentamers) at the expense of the larger aggregates presumably due to reductants in the cytosol. The observation that higher order aggregates are predominant does not detract from the manuscript, nor does it alter the conclusion that proinsulin aggregates are a not an artifact of cell lysis. The 2X lane does not appear very informative, as it doesn't support the author's view that the relative abundance of the proinsulin forms are similar (w/wo NEM) unless you ignore the large aggregates.

2) In human islets, the majority of proinsulin appears as dimers, rather than monomers even before treatment with SubAB (Figure 3D). Please explain.

3) The immunofluorescence microscopy figure (Figure 4C) showing islets from wild type and db/db mice could be improved by adding additional panels to solidify the author's conclusion that proinsulin accumulates in the ER of db/db, fed mouse islets. Specifically, it would be important to add a panel showing IF for proinsulin, calnexin and merge, as well as a second panel showing proinsulin, GM130 and merge for both the db/db fed and db/db fasted conditions. Preferably, these additional images would be incorporated into the main-text figure (Figure 4C), perhaps replacing the second panels that are currently shown for the fed and fasted conditions. Alternatively, these panels could be incorporated into the supplemental figure (Figure S3) that shows the quantitative analysis of proinsulin/calnexin colocalization for the db/db mice, fed and fasted conditions.

4) A central feature of the author's model is that the accumulation of disulfide-linked proinsulin aggregates is an early event in T2D that causes activation of the UPR pathway. The reviewers concluded that it would be important to document increased ER stress in islets from the db/db mice at the same stages of T2D progression that were analyzed for insulin expression and proinsulin aggregate formation (Figure 4D). Specifically, is BiP expression already elevated in islets from mice that have normal blood glucose levels, but >90% proinsulin aggregates. BiP expression could either be evaluated by immunofluoresence microscopy, or by protein immunoblotting.

5) The manuscript is written as if it is targeted for diabetes experts. Better background with logic developed should be provided for certain tools such as the Akita mouse and Leptin receptor mutant line to enhance readability by a wider audience.

---

## [Author Response]

Essential revisions:1) The majority of data is based on non-reducing SDS-PAGE immunoblots. The authors performed multiple experiments to demonstrate the specificity of the assay. The authors should reconsider their interpretation of the experiments shown in Figure 2C. The NEM treatment prior to cell lysis (lane 2) seems to stabilize very large aggregates (>pentamers). Cell lysis without blocking free thiols favors formation of the proinsulin ladders (dimers to pentamers) at the expense of the larger aggregates presumably due to reductants in the cytosol. The observation that higher order aggregates are predominant does not detract from the manuscript, nor does it alter the conclusion that proinsulin aggregates are a not an artifact of cell lysis. The 2X lane does not appear very informative, as it doesn't support the author's view that the relative abundance of the proinsulin forms are similar (w/wo NEM) unless you ignore the large aggregates.

We fully concur with the reviewers’ observation that (1) omission of NEM treatment of the cells/lysate favors detection of proinsulin ladders (monomers to pentamers) whereas (2) pretreatment of cells with NEM before (and during) cell lysis results in the detection of dramatically less of the ladder of disulfide-linked proinsulin oligomers (Figure 2C) — and additionally, NEM pretreatment favors an increase in the detection of higher molecular weight proinsulin-containing complexes. We have more explicitly stated this point in the Results, consistent with the reviewers’ comments. Additionally, as requested, we have removed the overexposure lane “2X” (and altered the Figure Legend accordingly).

To continue with this point, there are two alternative possibilities that the reviewers seek to bring to our attention: our hypothesis that upon PERK inhibitor treatment or other disorders of ER homeostasis, disulfidelinked proinsulin oligomers are abundant in the cells but their detection with our antibodies is inhibited by alkylation with NEM, and the reviewers’ alternative that proinsulin oligomers are formed by dissolution of higher molecular weight complexes and it is their formation that is inhibited (by consumption of putative cytosolic reductants) upon NEM treatment. The reviewers have asked that we give further consideration to this sentence from our main text, which said: “The *relative* abundance of the distinct disulfide-linked *oligomeric species* of proinsulin was essentially unchanged in cells that were either alkylated *in situ* with 20 mM N-ethyl maleimide prior to cell lysis, or not (Figure 2C)“. Our sentence did not address the higher molecular weight complexes but rather the relative abundance of monomers, dimers, trimers, etc., so our sentence was actually correct as written. But the reviewers’ comment prompted us to consider the alternatives proposed above, which led us to perform new experiments, described below.

First, when we treat cells with NEM, there is indeed an increase in the highest molecular weight proinsulin complexes, but it is sufficiently modest that on a light exposure like that shown in Author response image 1 (Panel A), the band intensity increase of the higher molecular weight complexes is almost undetectable, and does not account for the loss of signal in the lower molecular weight ladder of disulfide-linked proinsulin bands. Second, in Author response image 1 (Panel B), identical wells of INS1 β cells incubated O/N with PERK inhibitor were treated one of three ways: (1) we allowed the lysate to remain on ice for one hour before adding sample buffer and running the gel (first and third lanes) – we have checked this previously and can confirm that this has no impact on the appearance of the ladder of disulfide-linked proinsulin bands. (2) we pre-treated cells with NEM before (and during) lysis (second lane), exactly as done in the manuscript Figure 2C which attracted the reviewers’ comment, or (3) we allowed the lysate to remain on ice for one hour, exactly as in lane 1, and only afterwards treated the lysate with 2 mM NEM (last lane, marked with asterisk).

Then all were resolved by nonreducing SDS-PAGE (moreover, we always treat the nonreducing gel, after it has finished running, with 25 mM DTT to reduce all possible disulfide complexes prior to electrotransfer and immunoblotting). NEM pretreatment before lysis caused a drastic loss of the detection of the proinsulin ladder of bands (second lane), consistent with the reviewers’ comment. However, initiating NEM treatment of lysates 1 hour post-lysis yielded a similar loss of detection of the ladder of disulfide-linked proinsulin oligomeric bands (panel B, lane 4) (by reducing SDS-PAGE the reduced monomeric proinsulin band is comparably recovered in all samples, as in panel A). This result has been repeated and confirmed x 3. The cytosolic volume of a cell is on the order of 1 picoliter, so at the time of cell lysis, reductants of the cytosol (glutathione, etc.) are suddenly released but thereafter are drastically diluted. If the lower molecular weight ladder of disulfide-linked proinsulin bands is formed by dissolution of higher molecular weight complexes at the time of lysis in the absence of NEM, then this should have happened in the latter condition; however, this was not observed. We’ve also obtained the very same result by treating with 2 mM NEM after SDS gel-sample buffer has already been mixed with the lysate sample. To us, these data suggest that alkylation impairs detection of the lower molecular weight ladder of bands with our antibodies. We don’t seek to exclude the reviewers’ point of view, indeed, we have not elaborated on this point in the manuscript – but we want the reviewers to understand our thought process. Additionally, we strongly agree with the reviewers and thank them for their point that the finding of higher order aggregates (which are actually enriched in live cells treated with NEM or treated with menadione — as shown in a companion manuscript) does not alter our conclusion that disulfidelinked proinsulin complexes are not an artifact of cell lysis.

2) In human islets, the majority of proinsulin appears as dimers, rather than monomers even before treatment with SubAB (Figure 3D). Please explain.

The reviewers’ point is well taken. Unlike our work in congenic mice or β cell lines which have a smaller range of variability between samples, the results of islets isolated and shipped from genetically (and environmentally) unrelated human donors do vary meaningfully, so we must not assume that an identical result should be obtained for all humans. We have now analyzed a few additional human islet preps. In the nonreducing SDS-PAGE shown in Author response image 2 (panel A) at right, human islet prep #1 reveals proinsulin monomers, which are nearly undetectable in islet prep #2 from an unrelated human donor. Similarly, in Author response image 2 (panel B) we obtained a human islet prep in which proinsulin monomer was the primarily species detected, until the islets were incubated overnight with PERK inhibitor. These experiments have now been added as Figure 2—figure supplement 1 and Figure 3—figure supplement 1. We also note that in rodent β cells, covalent proinsulin dimers were routinely detected (see Figure 1B of the manuscript). The proinsulin disulfidelinked complexes are a biomarker of proinsulin misfolding; however, whereas we prepare rodent islets (or β cell lines) ourselves, we obtain human islets from external sources and we do not control clinical conditions at the time of obtaining the pancreas (needless to say, no healthy living person willingly gives up their pancreas!). Moreover, even when the clinical record of human patients shows no evidence of diabetes, that alone does not assure metabolic uniformity between different individuals, nor does it address any technical differences between human islet preparations. All in all, this does not alter our conclusion that the recovery of proinsulin covalent dimers from the islets of non-diabetic humans strongly suggests that proinsulin misfolding can occur in normal β cells without *INS* gene mutations, and this can occur prior to the detectable onset of diabetes.

**Author response image 2. respfig2:** 

3) The immunofluorescence microscopy figure (Figure 4C) showing islets from wild type and db/db mice could be improved by adding additional panels to solidify the author's conclusion that proinsulin accumulates in the ER of db/db, fed mouse islets. Specifically, it would be important to add a panel showing IF for proinsulin, calnexin and merge, as well as a second panel showing proinsulin, GM130 and merge for both the db/db fed and db/db fasted conditions. Preferably, these additional images would be incorporated into the main-text figure (Figure 4C), perhaps replacing the second panels that are currently shown for the fed and fasted conditions. Alternatively, these panels could be incorporated into the supplemental figure (Figure 4—figure supplement 1and Figure 8) that shows the quantitative analysis of proinsulin/calnexin colocalization for the db/db mice, fed and fasted conditions.

Thanks to the reviewers for this suggestion; we have followed the advice and added a new Figure 5—figure supplement 1 containing additional images including those suggested by the reviewers and additional comparators (note that we have changed the numbering of the Supplemental figures by instructions from the Journal, but we have not deleted any data from the manuscript).

4) A central feature of the author's model is that the accumulation of disulfide-linked proinsulin aggregates is an early event in T2D that causes activation of the UPR pathway. The reviewers concluded that it would be important to document increased ER stress in islets from the db/db mice at the same stages of T2D progression that were analyzed for insulin expression and proinsulin aggregate formation (Figure 4D). Specifically, is BiP expression already elevated in islets from mice that have normal blood glucose levels, but >90% proinsulin aggregates. BiP expression could either be evaluated by immunofluoresence microscopy, or by protein immunoblotting.

We appreciate the suggestion, thanks. As the reviewers may know, in the paper we cited about human islets (Marchetti et al., 2007), neither BiP, XBP1, or CHOP were higher in T2D β cells or isolated islets cultured at normal glucose. In *db/db* mice. (Laybutt et al., 2007) reported that one of the most robust ER stress response markers was the target gene DNAJC3 encoding p58ipk (also known as ERdj6). p58ipk is a cochaperone for BiP that has been studied in many labs including those of David Ron, Linda Hendershot, Markus Stoffel, Warren Ladiges, in addition to one of our own team (RJK) as well as others. The stages of T2D progression were presented in the previous Figure 5. We performed new experiments and introduced these results as new Figure 5A and 5B; the previous Figure 5 is now Figure 5C. The results are in general agreement with those reported by Ross Laybutt, Trevor Biden and others.

5) The manuscript is written as if it is targeted for diabetes experts. Better background with logic developed should be provided for certain tools such as the Akita mouse and Leptin receptor mutant line to enhance readability by a wider audience.

In response to this concern, we have changed words in the Abstract and Introduction to “leptin receptordeficient animals” and clarified that such animals are predisposed to hyperphagia (we used the word “overeating” that anyone can understand). Further, in the Introduction, we have modified the paragraph first referring to the *Akita* mouse, as follows: “Insulin-deficiency caused directly by proinsulin misfolding has been proved unequivocally in an autosomal-dominant form of diabetes known as Mutant *INS*-gene-induced Diabetes of Youth (MIDY) (Liu et al., 2010b), which occurs in patients bearing one mutant and one wild-type *INS* allele (Liu et al., 2015; Stoy et al., 2010). The disease in humans is pathogenetically identical to that seen in the mutant *Akita* diabetic mouse (Izumi et al., 2003) or Munich MIDY Pig (Blutke et al., 2017) – which are animals expressing one mutant *INS* allele encoding proinsulin-C(A7)Y that is quantitatively misfolded due to an inability to form the Cys(B7)-Cys(A7) disulfide bond. Ordinarily the expression of only one WT *INS* allele would be sufficient to avoid diabetes, but *Akita* mice develop diabetes despite expressing three alleles encoding WT proinsulin in addition to the one encoding mutant proinsulin (Liu et al., 2010b). Both preclinical and clinical data prove that in MIDY, it is the expression of misfolded proinsulin that triggers diabetes…”